
# Airborne measurements of directional reflectivity over the marginal sea ice zone

Sebastian Becker[1], André Ehrlich[1], Evelyn Jäkel[1], Tim Carlsen[2], Michael Schäfer[1], and Manfred Wendisch[1]

[1]Leipzig Institute for Meteorology (LIM), Universität Leipzig, Leipzig, Germany
[2]Department of Geosciences, University of Oslo, Oslo, Norway

**Correspondence:** Sebastian Becker (sebastian.becker@uni-leipzig.de)

**Abstract.** The directional reflection of solar radiation by the Arctic Ocean is dominated by two main surface types: sea ice (often snow-covered) and ice-free (open) ocean. However, in the transitional marginal sea ice zone (MIZ), the reflection properties of both surface types are mixed, which might cause uncertainties in the results of retrieval methods of atmospheric parameters over the MIZ using airborne and satellite measurements. To quantify these uncertainties, respective measurements of reflection

properties of the MIZ are needed. Therefore, in this study, an averaged hemispherical-directional reflectance factor (HDRF) of the inhomogeneous surface (mixture of sea ice and open ocean surfaces) in the MIZ is derived using airborne measurements collected with a digital fish-eye camera. For this purpose, a sea ice mask was constructed to separate the reflectivity measurements from sea ice and open ocean pixels. The separated data sets were accumulated and averaged to provide separate HDRFs for sea ice and open ocean surfaces. The respective results were compared with simulations and independent measurements

available from the literature. Using the sea ice fraction derived in parallel from the digital camera images, the mixed HDRF describing the directional reflectivity of the inhomogeneous surface of the MIZ was reconstructed by a linear weighting procedure. The result was compared with the original measurements of directional reflectivity over the MIZ. It is concluded that the HDRF of the MIZ can be well reconstructed by linear combination of the HDRFs of homogeneous sea ice and open ocean surfaces, accounting for the special conditions present in the MIZ compared to homogeneous surfaces.

# 1 Introduction

The Arctic Ocean is a key element of the complex Arctic climate system. From a solar radiative point of view it is characterized by a strong contrast of bright sea ice surfaces and dark areas of open ocean, which both determine the surface solar radiative energy budget. The reflection properties of both surface types are strongly linked to the sea ice-albedo-feedback,

which significantly contributes to Arctic amplification (Wendisch et al., 2017; Pithan and Mauritsen, 2014). Especially the high spectral albedo of the snow-covered Arctic sea ice, which typically ranges between 0.8 and 0.9 in the visible spectral range (Wiscombe, 1980), increases the reflected solar radiation. The combined snow-sea ice albedo depends mainly on snow





cover, grain size, melt pond fraction, and solar zenith angle (SZA). In contrast, more solar radiation is absorbed by open ocean. Its broadband albedo depends on surface wind speed and SZA, and typically ranges below 0.1 for SZAs lower than 65° in

clear-sky conditions (Jin et al., 2004; Feng et al., 2016).

The reflection of solar radiation by snow-covered sea ice and open ocean surfaces was extensively studied and characterized by ground-based, airborne, and satellite observations (e. g., Cox and Munk, 1954; Gatebe et al., 2003; Bourgeois et al., 2006). However, large areas of the Arctic ocean are characterized by a mixture of sea ice and open ocean, e. g., in case of leads, polynyas, or melt ponds (e. g., Hoffman et al., 2019). Also, in the marginal sea ice zone (MIZ), individual ice floes of different

sizes cover varying fractions of the ocean. Strong and Rigor (2013) defined the MIZ as the zone where the sea ice concentration ranges between 0.15 and 0.8. In a warmer Arctic, the sea ice cover will retreat further (Perovich et al., 2018) and first year sea ice becomes more dominant. Accordingly, a significant reduction of the sea ice thickness was observed (Kwok, 2018). In summer, the thinner sea ice is more dynamic and breaks more easily, which leads to a higher amount of leads, a stronger drift, a faster dispersion of the floes, and a more expanded MIZ (Kashiwase et al., 2017). Strong and Rigor (2013) showed

that the width of the MIZ increased by about 13 km per decade in summer resulting in a widening of 39 % between 1979 and 2011. Accordingly, the summertime MIZ nowadays covers between 20 % and 60 % of the entire Arctic sea ice extent (Rolph et al., 2020). In winter, when the MIZ is characterized by freshly frozen sea ice, no significant trend was observed. Therefore, a characterization of the radiative properties of the mixture of sea ice and open ocean in the MIZ is needed to better quantify the complex radiative processes in this area.

To quantify the solar radiative energy budget at the surface, the surface albedo as a hemispherically integrated measure of reflection is sufficient. However, satellites measure the directional reflection instead of the surface albedo, such that conversion methods quantifying the reflection characteristics of a surface in all directions need to be applied (e. g., Schaaf et al., 2002). The directional distribution of the reflected radiation (as fraction of the incident radiation) is often described by the bidirectional reflectance distribution function (BRDF, mostly used in models) or the directly measurable hemispherical-directional

reflectance factor (HDRF; Nicodemus et al., 1977; Schaepman-Strub et al., 2006).

The HDRF of homogeneous snow/sea ice and open ocean surfaces has been derived from measurements and simulations in numerous studies (e. g., Bourgeois et al., 2006; Gatebe et al., 2003). The HDRF of snow is characterized by high reflectance, which is further enhanced in forward direction (i.e., at high reflection zenith angles around 0° relative azimuth). This forward peak increases relative to the reflectance in nadir direction with increasing solar zenith angle and wavelength (e. g., Warren

et al., 1998; Aoki et al., 2000; Carlsen et al., 2020). Additionally, the snow HDRF depends on snow morphology (i. e., snow grain size and shape), which is crucial for accurate simulations of the snow HDRF (Jafariserajehlou et al., 2021). In contrast, the open ocean is characterized by low reflectance because of the strong absorption of solar radiation by the liquid water. However, in the so called sunglint angular range, specular reflection can be several orders of magnitudes higher. The width and maximum intensity of the sunglint area depend on surface roughness and, thus, on the wind speed. The higher the wind speed,

the rougher the surface and the broader the sunglint area but the lower its maximum intensity (e. g., Cox and Munk, 1954; Jackson and Alpers, 2010).





Quantitative measurements of the HDRF of snow and open ocean were obtained by ground-based, airborne, and satellites observations. Bourgeois et al. (2006) and Marks et al. (2015) retrieved the HDRF of snow using a ground-based goniospectrom-eter, which measures the spectral radiance reflected from a small surface area from different directions. Their measurements

provide an angular resolution of about 15° with a spectral resolution of about 1 nm. Commercial digital cameras equipped with a 180° fish-eye lens provide instantaneous reflectance observations as a function of reflection directions of the lower hemisphere with high angular resolution. Goyens et al. (2018) combined ground-based goniometer and fish-eye camera ob-servations to benefit from both the hyperspectral resolution and the hyperangular resolution. Since goniometers are restricted to observations over solid ground, mostly shipborne or airborne measurements are used to derive the HDRF of open ocean

(Cox and Munk, 1954). Gatebe et al. (2005) retrieved the open ocean wind speed and BDRF using the Cloud Absorption Radiometer (CAR), a scanning radiometer with a field of view (FOV) of 190°. This wide FOV is achieved by scanning the lower hemisphere with a narrow FOV optic of about 1° using a scanning mirror.

Airborne observations with a digital camera equipped with a wide-angle lens have been reported by Ehrlich et al. (2012). Although the spectral resolution of the camera is limited (only the three visible channels red, green and blue are available), this

method was applied to analyze the impact of surface winds on the HDRF of the ocean surface. For homogeneous snow-covered areas in the Antarctic, Carlsen et al. (2020) used airborne 180° fish-eye camera observations to quantify the anisotropy of the snow HDRF with changing surface roughness, snow grain size, and SZA.

Most of the ground-based and airborne BRDF studies published so far focused on a specific homogeneous surface type. The reflectivity of an inhomogeneous mixture of different surface types, such as present in the MIZ, has rarely been observed in

the past. Qu et al. (2016) linearly combined simulated BRDFs of the different surface types in the MIZ in order to retrieve the surface albedo in the MIZ from satellite observations. This approach is limited, since the open ocean surfaces are inhomoge-neously distributed between the ice floes in the MIZ. Thus, waves are attenuated (Kohout et al., 2011) and the dependence of the sunglint on the surface roughness and the wind speed could differ from ice-free open oceans. Therefore, in our study, the spatial coverage of airborne observations with a 180° fish-eye camera is used to characterize the HDRF of a mixture of open

ocean and snow-covered sea ice surfaces in the MIZ. The observations have been obtained during the ACLOUD (Arctic CLoud Observations Using airborne measurements during polar Day) campaign northwest of Svalbard (Wendisch et al., 2019). The directional reflectance quantities as well as the observations are introduced in Sect. 2. In Sect. 3, the separation of the observed mixed scenes into the contributions of sea ice and open ocean is described by applying a sea ice mask. This allows the analysis of the individual HDRFs of sea ice and open ocean in the MIZ, which are presented in Sect. 4 together with a comparison to

HDRFs of homogeneous surfaces. Finally, the separated HDRFs are recombined for different sea ice fractions and compared to the original observations in Sect. 5.



## 2 Methodology and measurements

### 2.1 Definition of reflectance quantities

The spectral BRDF $f_{\mathrm{BRDF}}$ of a surface describes the directional distribution of the reflected radiation and is defined as:

$$f_{\mathrm{BRDF}}(\theta_{\mathrm{i}}, \phi_{\mathrm{i}}; \theta_{\mathrm{r}}, \phi_{\mathrm{r}}, \lambda) = \frac{\mathrm{d}I_{\mathrm{r}}(\theta_{\mathrm{i}}, \phi_{\mathrm{i}}; \theta_{\mathrm{r}}, \phi_{\mathrm{r}}, \lambda)}{\mathrm{d}F_{\mathrm{i}}(\theta_{\mathrm{i}}, \phi_{\mathrm{i}}, \lambda)} \tag{1}$$

(Nicodemus et al., 1977) . $F_{\mathrm{i}}$ represents the spectral irradiance (in $\mathrm{W\,m^{-2}\,nm^{-1}}$) illuminating (subscript "i") a surface at wavelength $\lambda$ from the direction characterized by the incident zenith and azimuth angles, $\theta_{\mathrm{i}}$ and $\phi_{\mathrm{i}}$, respectively. $I_{\mathrm{r}}$ quantifies the radiance (in $\mathrm{W\,m^{-2}\,nm^{-1}\,sr^{-1}}$) reflected (subscript "r") into the direction characterized by the reflection zenith and azimuth angles, $\theta_{\mathrm{r}}$ and $\phi_{\mathrm{r}}$, respectively, and depends additionally on the incident angles. The BRDF has the unit of inverse steradiant ($\mathrm{sr^{-1}}$). Often the bidirectional reflectance factor (BRF) $R_{\mathrm{BRF}}$ is used instead of the BRDF. The reflectance factor (without unit) is defined as the ratio of the BRDF of the actual surface to the constant BRDF of a Lambertian surface, which is equal to $1/\pi\,\mathrm{sr^{-1}}$. Thus:

$$R_{\mathrm{BRF}} = \pi\,\mathrm{sr} \cdot f_{\mathrm{BRDF}}. \tag{2}$$

Since the illumination under atmospheric conditions is a combination of a direct and a hemispherical diffuse irradiance component, $F_{\mathrm{dir}}$ and $F_{\mathrm{diff}}$, respectively, both BRDF and BRF cannot be measured practically. Therefore, the hemispherical-directional reflectance factor (HDRF, without unit) $R_{\mathrm{HDRF}}$ is introduced (e. g., Schaepman-Strub et al., 2006). However, if the diffuse fraction of the incident radiation is sufficiently small, the HDRF represents a good approximation of the BRF. The HDRF is determined by:

$$R_{\mathrm{HDRF}}(\theta_{\mathrm{i}}, \phi_{\mathrm{i}}, 2\pi; \theta_{\mathrm{r}}, \phi_{\mathrm{r}}) = \frac{\pi\,\mathrm{sr} \cdot I_{\mathrm{r}}(\theta_{\mathrm{i}}, \phi_{\mathrm{i}}, 2\pi; \theta_{\mathrm{r}}, \phi_{\mathrm{r}})}{F_{\mathrm{i}}(\theta_{\mathrm{i}}, \phi_{\mathrm{i}}, 2\pi)}. \tag{3}$$

Here, the additional argument $2\pi$ refers to the diffuse radiation incidence over the whole hemisphere while $\theta_{\mathrm{i}}$ and $\phi_{\mathrm{i}}$ refer to the direction of the direct component. The spectral dependence is omitted here.

### 2.2 Observations and instrumentation

#### 2.2.1 Airborne campaign

In this paper, data collected during the Arctic CLoud Observations Using airborne measurements during polar Day (ACLOUD; Wendisch et al., 2019) campaign, which took place in May and June 2017, are analyzed. During ACLOUD, the MIZ was located at about 80° N in the region north-west of Svalbard (Norway). 19 measurement flights were conducted with each of the two research aircraft *Polar 5* and *Polar 6* from Alfred Wegener Institute, Helmholtz Centre for Polar and Marine Research (AWI; Wesche et al., 2016). Both aircraft were equipped with downward-looking 180° fish-eye cameras measuring the upward directional radiances (Ehrlich et al., 2019; Jäkel et al., 2019). Additionally, on *Polar 5* the Spectral Modular Airborne Radiation measurement SysTem (SMART; Wendisch et al., 2001) was operated to measure the spectral downward solar irradiance.





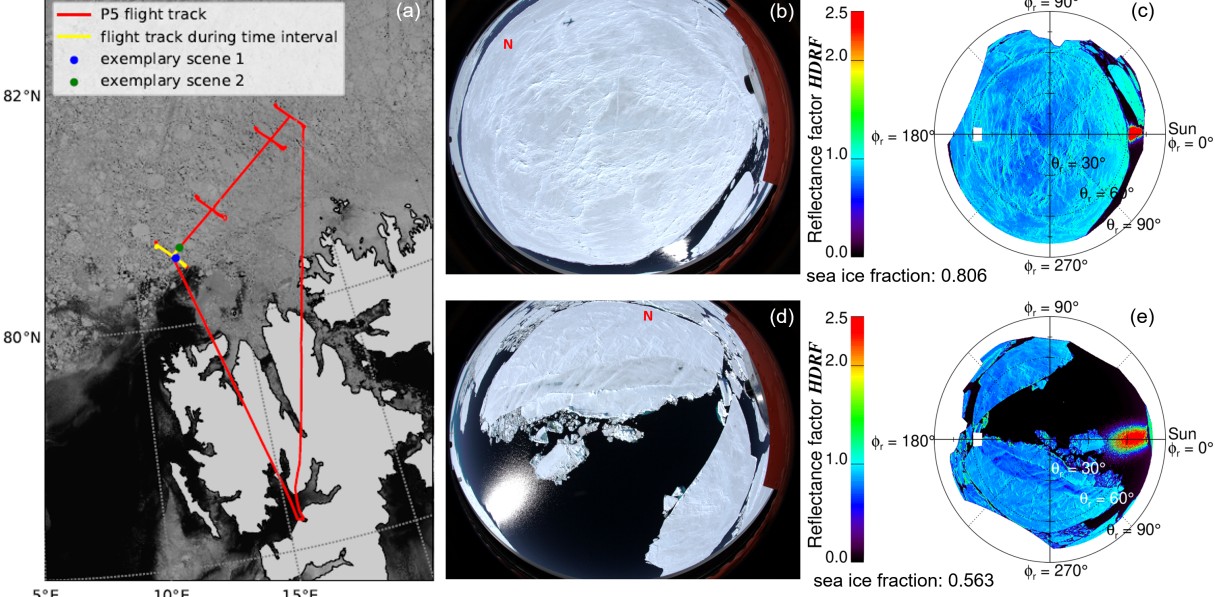

**Figure 1.** (a) Satellite image (composed of MODIS bands 1 and 2) of the MIZ north of Svalbard observed on 25 June 2017, 12:45 UTC including the flight track of *Polar 5* (red). The 20-minute period used for the analysis is highlighted in yellow. The blue and green dots point at the locations where the exemplary scenes in (b) and (d), resp., were observed. (b) and (d) show exemplary images taken by the fish-eye camera at 12:37:23 and 12:46:41 UTC, resp. (c) and (e) present the processed HDRF polar plots of the scenes shown in (b) and (d), resp. Note that the polar plots are rotated compared to the raw images in a manner that the solar incident direction is always to the right of the plot.

The data analyzed here were obtained during the flight on 25 June 2017 (flight number 23) performed under cloudless conditions. Fig. 1a shows a satellite image from the MODerate resolution Imaging Spectroradiometer (MODIS) instrument onboard the satellite Terra during the research flight. The surface was dominated by an inhomogeneous distribution of sea ice floes with diameters of up to several kilometers. The flight track of *Polar 5* is plotted in red in Fig. 1a. A 20-minute leg performed between 12:26:35 UTC and 12:46:47 UTC (highlighted in yellow in Fig. 1a) was selected for the analysis, which ensures stable environmental conditions. The solar zenith angle ranged between 57.7° and 58.0° with a mean of 57.8°. The observations were conducted along a 30 km straight flight section, during which the flight altitude varied between 65 and 165 m.

### 2.2.2   180° Fish-eye camera

The upward radiance was measured by a CANON EOS-1D Mark III digital camera equipped with a 180° fish-eye lens. Images with a resolution of 3906 × 2600 pixels were taken every 6 s. As common for commercial cameras, each pixel covers three spectral channels (RGB) centered at wavelengths of 591 nm (red), 530 nm (green), and 446 nm (blue) with a full width at half maximum (FWHM) of about 80 nm (Ehrlich et al., 2019; Carlsen et al., 2020). Figures 1b and 1d show two examples of raw true-color images taken by the fish-eye camera. The camera was calibrated in terms of geometrical, spectral and radiometric





characteristics, which allows a conversion of the measured raw data into radiances as described in detail by Carlsen et al. (2020).

In contrast to Carlsen et al. (2020), who applied a stellar method for the geometrical calibration, images of checkerboards taken from different perspectives served as reference. The images were analyzed by the open source routine `cv2.fisheye` from the free programming library OpenCV (http://opencv.org; Jiang, 2017). The backward model described by Urquhart et al. (2016) was applied to calculate the camera-fixed viewing zenith and azimuth angles, $\theta_\mathrm{v}$ and $\phi_\mathrm{v}$, respectively, of each image pixel using the OpenCV output parameters.

The fish-eye camera was fixed to the aircraft frame. To obtain radiance measurements with respect to an Earth-fixed coordinate system, the viewing angles $(\theta_\mathrm{v}, \phi_\mathrm{v})$ of each camera pixel were corrected to consider the aircraft attitude angles (roll, pitch, yaw). Euler rotation matrices were applied to transform the viewing angles into the reflection angles $(\theta_\mathrm{r}$ and $\phi_\mathrm{r})$ (Ehrlich et al., 2012). The azimuth plane of the images was rotated with respect to the relative position of the Sun, such that the Sun (and the forward direction) is located at the right in all polar plots shown in this paper. The footprint of one single image varied between

380 m and 915 m for the altitude range of the 20-minute leg assuming that the effective FOV of the fish-eye lens is 160°.

### 2.2.3 Calculation and uncertainty of the HDRF

Combining the downward irradiance $F_\mathrm{i}$ measured by SMART (Jäkel et al., 2019) and the angularly resolved radiances $I_\mathrm{r}$ from the fish-eye camera (Jäkel and Ehrlich, 2019) allows the calculation of the HDRF in flight altitude (Eq. 3), whereby the spectrally resolved irradiances were converted into the spectral range of each camera channel using the individual relative

spectral response function from the spectral calibration.

The uncertainty of the radiometric calibration of the fish-eye camera was estimated by 4 % (Carlsen et al., 2020). Further uncertainties stem from the sensor characteristics, the geometric calibration, and the aircraft attitude correction, leading to a total uncertainty of the fish-eye camera radiance measurements of about 4.2 %. However, during ACLOUD complementing radiance measurements were performed with SMART and a spectral imager, which demonstrated deviations of up to 35 % for

the blue channel, while reflected radiances measured in the green and red channels ranged within the measurement uncertainties (Ehrlich et al., 2019). As a consequence, the fish-eye camera was inter-calibrated. However, for the further analysis in this study only radiances measured in the red channel were selected, since they have shown the best agreement within the instrumental intercomparison. The root mean square deviation between the radiances of the digital camera and SMART in the red channel amounts to $0.01\,\mathrm{W\,m^{-2}\,nm^{-1}\,sr^{-1}}$ and the correlation coefficient is 0.98.

According to Bierwirth et al. (2009), the total uncertainty of the SMART irradiance measurements is 3.2 % in the visible spectral range. However, an updated transfer calibration (3 % error) and a larger cosine correction error (2 %) due to the larger solar zenith angle lead to an increased total uncertainty of 4.3 %. Thus, the total uncertainty of the calculated HDRF amounts to about 6 %.

Radiative transfer simulations performed with the library for radiative transfer (libRadtran; Emde et al., 2016) were used to

estimate the impact of the atmosphere between the ground and the maximum flight altitude (165 m) on the measured HDRF. They have shown, that the difference between the HDRF at ground and flight level is less than 1 % for the red camera channel.





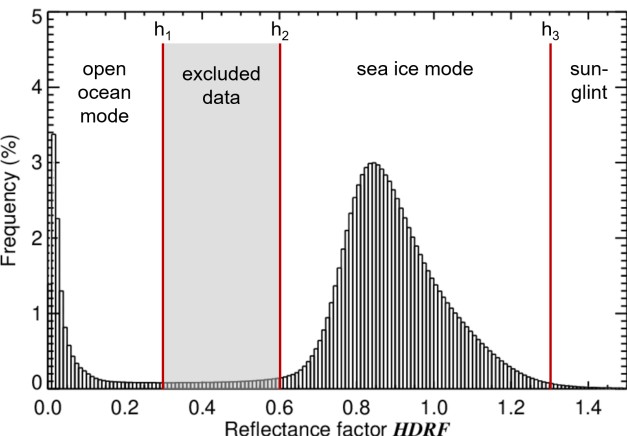

**Figure 2.** Frequency distribution of the observed HDRF for all directions and all images. The three vertical red lines indicate the thresholds of $h_1 = 0.3$, $h_2 = 0.6$ and $h_3 = 1.3$ applied in the sea ice mask algorithm.

The fully calibrated and processed HDRF obtained from the exemplary raw images in Figs. 1b and 1d are shown in Figs. 1c and 1e, respectively. The region contaminated by the shadow of the aircraft ($\theta_r \approx 60°$, $\phi_r \approx 180°$) was excluded and is represented by the white gap in the polar plots. The HDRF plots of the exemplary scenes reveal a high contrast between sea ice and open ocean areas and show a largely enhanced HDRF in the sunglint region of open ocean.

## 3 Separation of sea ice and open ocean surfaces

### 3.1 Sea ice mask

138 fish-eye camera images taken within the 20-minute period were analyzed. Most of the individual images show a mixture of sea ice and open ocean. To obtain separate HDRFs for either surface type, a sea ice mask is constructed, which considers the different reflection characteristics of both surface types. Figure 2 shows a frequency distribution of all HDRF measurements, merging all images and all directions (pixels). The histogram reveals a distinct separation of the data into two modes, originating from open ocean and sea ice. Accordingly, all pixels with HDRF values below a threshold of $h_1 = 0.3$ are assigned to open ocean, pixels with HDRF values above a second threshold of $h_2 = 0.6$ are mostly related to sea ice but can also be assigned to open ocean in case of specular reflection observations. HDRF values between both thresholds, which are mostly linked to the ice floe edges, amount to roughly 3 % of the data and were excluded from further analysis.

In the sunglint region, also the HDRF of open ocean can feature values similar to or even exceeding typical values of sea ice. This sunglint area was identified by two additional criteria. Firstly, HDRF values larger than a third threshold of $h_3 = 1.3$ were considered as sunglint and assigned to open ocean. Secondly, a color ratio defined by the ratio of the radiances measured in the red and the blue camera channel was used to identify the edges of the sunglint zone. Since the sunglint area appears more yellowish than the sea ice, its color ratio is higher. A color ratio threshold of $c = 0.95$ was chosen for the separation of sea ice





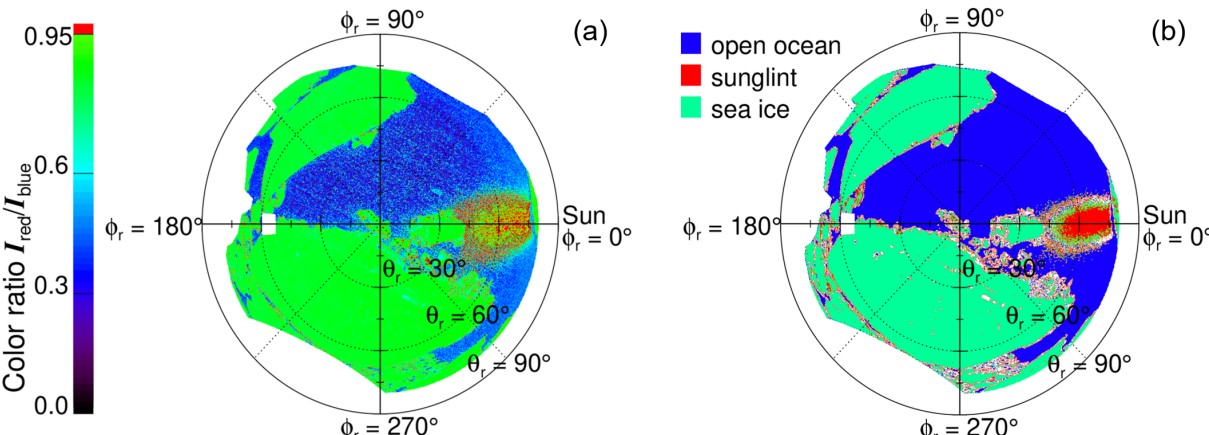

**Figure 3.** (a) Color ratio of an exemplary scene (same scene as in Fig. 1d). The area identified as sunglint (above the color ratio threshold $c = 0.95$) is highlighted in red. (b) Allocation of the pixels to sea ice (turquoise), open ocean (blue) or the sunglint area of open ocean (red) applying the sea ice mask illustrated in Fig. 4.

and sunglint when the HDRF was between $h_2$ and $h_3$. In Fig. 3a the color ratio of the exemplary scene (same as in Fig. 1d) is shown with values higher than 0.95 highlighted in red. Together with Fig. 3b, which shows the surface types identified by the sea ice mask, it demonstrates that the mask is capable to separate sea ice, open ocean, and sunglint. The complete decision process of the sea ice mask is summarized in Fig. 4. An uncertainty analysis of the sea ice mask with respect to the applied

thresholds is given in the following section based on the calculated sea ice fraction.

## 3.2 Sea ice fraction

Using the sea ice mask, the sea ice fraction was calculated for each image. The sea ice fraction usually refers to a horizontal surface. The fish-eye lens of the camera, however, weighs the individual pixels equally although each pixel refers to a different surface area. Nadir pixels cover a smaller area than pixels close to the horizon. These different pixel projections were taken

into account when the sea ice fraction averaged over a whole image was calculated.

Within the analyzed time interval, the sea ice fraction varied between 0.35 and 1.0. Fig. 5 shows the frequency distribution of the observed sea ice fractions. Images with higher sea ice fractions were more frequent than images dominated by open ocean. The mean sea ice fraction of the area sampled during the 20-minute measurement time interval amounts to 0.83 (blue line in Fig. 5). The accuracy of the sea ice fraction depends on the choice of the thresholds that are applied in the sea ice mask. In

order to estimate the uncertainty related to the choice of the HDRF threshold values, a sensitivity study was performed, slightly varying one of the thresholds while the others were held constant.

The thresholds $h_1$ and $h_2$ were varied between the two modes (0.2 to 0.6). Changing $h_1$ or $h_2$ by 0.1 leads to a change in sea ice fraction of about 1.2 %. $h_3$ was decreased down to 1.1. The color ratio threshold $c$ was varied between 0.9 (lower values clearly misclassify sea ice as sunglint) and 1.0, whereby the sensitivity of the sea ice fraction is larger when $c$ is decreased. The





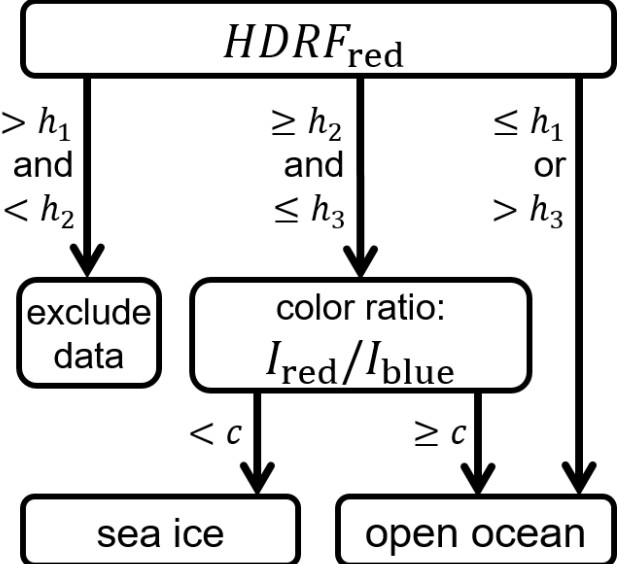

**Figure 4.** Decision tree separating pixels covered by sea ice and open ocean defining the sea ice mask. The applied thresholds $h_1 = 0.3$, $h_2 = 0.6$ and $h_3 = 1.3$ are based on the HDRF retrieved for the red camera channel $HDRF_{\mathrm{red}}$. The threshold $c = 0.95$ is based on the ratio of the radiances measured in the red $I_{\mathrm{red}}$ and the blue $I_{\mathrm{blue}}$ channel.

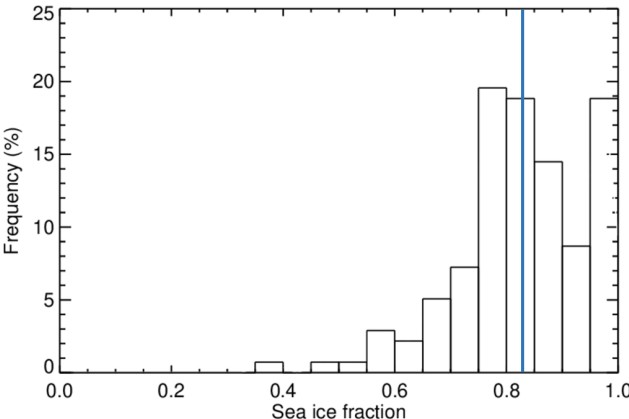

**Figure 5.** Frequency distribution of the sea ice fraction for all images taken within the 20-minute time interval, the vertical blue line represents the mean sea ice fraction of 0.83.

averaged sea ice fraction resulting from the variation of the thresholds ranges between 0.72 and 0.86. Since the misclassification of pixels is obvious for extreme cases, the uncertainty of the sea ice fraction due to the sea ice mask is estimated to be less than 4 %.





## 4 Separated HDRF and comparison with simulations and observations from the literature

After applying the sea ice mask to each of the images, all HDRF measurements of one single direction (pixel) assigned to

either sea ice or open ocean were averaged. Doing so for all directions leads to a separated HDRF for each of the two surface types.

### 4.1 HDRF of open ocean

Figure 6a shows the average HDRF of the open ocean areas separated from the observations in the MIZ (separated open ocean HDRF). For most reflection directions the HDRF values are below 0.3. The average HDRF outside the sunglint region is

0.11 with an uncertainty of $\pm 0.02$ when different thresholds are applied in the sea ice mask algorithm. In the sunglint region $(\theta_\mathrm{r} \approx 60°, \phi_\mathrm{r} \approx 0°)$ the reflection is significantly enhanced and exceeds the maximum of the scale chosen here. A cross-section of the separated open ocean HDRF along the solar principle plane is illustrated in Fig. 6b (blue line) and shows the full dynamic range of the sunglint with a maximum HDRF of around 9. Both the polar plot (Fig. 6a) and the cross-section (Fig. 6b) reveal a highly variable pattern of the entire separated open ocean HDRF. The standard deviation (blue shading in Fig. 6b) is up to 0.6

outside the sunglint and up to 9.2 in the sunglint region. The high variability is a result of the high sea ice fraction, which leads to a low number of open ocean pixels used for averaging. The sea ice fraction for each direction (pixel) of the solar principal plane (calculated as ratio of all images) is referred to as pixel-based sea ice fraction in the following and is indicated by the black line in Fig. 6b. Although there is also some directional variability in the pixel-based sea ice fraction, it is higher than 0.8 for most reflection directions.

In the observations presented here, the sunglint does not have the shape of a typical Gaussian distribution as reported in literature (e. g., Cox and Munk, 1954; Gatebe and King, 2016; Ehrlich et al., 2012). Instead, several smaller local maxima are obvious beside a global one and the sunglint is slurred towards the horizon. The irregular shape of the sunglint is likely a result of the low amount of observations and is imprinted in the pixel-based sunglint fraction (grey line in Fig. 6b). The pixel-based sunglint fraction is defined as the ratio of the number of observations identified as sunglint to the number of all

observations assigned to open ocean by the sea ice mask for the respective direction. It implies that the shape of the sunglint is highly variable among the images used for averaging, ranging from pure specular reflection of the Sun to sunglints cut by the edge of an ice floe (i. e., Fig. 1b) and widely blurred sunglints (i. e., Fig. 1d). The width of the sunglint primarily depends on the surface roughness, which is related to the surface wind speed (Cox and Munk, 1954). However, the distribution of sea ice and open ocean in the MIZ affects the surface roughness and its dependence on the wind speed. The development of waves is

weaker in the gaps between ice floes than in the homogeneous open ocean (Kohout et al., 2011). This leads to the hypothesis that the shape, extent and also position of the sunglint in the MIZ largely depend on the size of the open ocean areas between the ice floes. The ice floe distribution differed significantly between the images, which might have caused the diversity of the sunglints observed in these measurements.

The dependence of wind speed and sunglint was tested by comparing the separated open ocean HDRF to radiative transfer

simulations. The simulated open ocean HDRF was obtained from libRadtran, which incorporates BRDF parametrizations


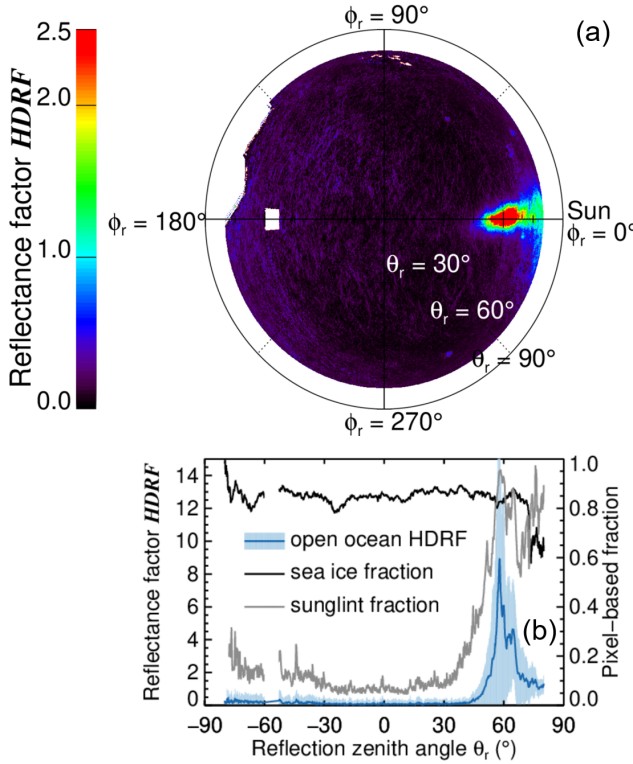

**Figure 6.** (a) Polar plot of the separated mean HDRF of open ocean. (b) Cross-section of the HDRF of open ocean along the solar principal plane (blue). The shaded area shows the standard deviation calculated for each direction of the solar principal plane from the entire time series (138 images). The black and grey lines denote the mean pixel-based sea ice fraction and the mean pixel-based sunglint fraction, resp. (second y-axis).

of different surface types (Mayer and Kylling, 2005) including open ocean (Cox and Munk, 1954). The simulations were performed for different wind speeds ranging from $1\,\mathrm{m\,s^{-1}}$ (minimum possible wind speed in libRadtran) to $10\,\mathrm{m\,s^{-1}}$. The atmospheric conditions were provided by measurements of a radiosonde launched at Ny-Ålesund at 12 UTC on 25 June 2017. Above, the default sub-Arctic summer atmosphere was selected. The solar zenith angle was set to $57.8°$, corresponding to the

observation time. The hemispherical downward irradiance and the directional upward radiances at ground level (resolution $3°$ in zenith and $5°$ in azimuth) were simulated in a wavelength range between 350 and 750 nm and were converted into the spectral range of the red camera channel.

Cross-sections of the simulated open ocean HDRF for several wind speeds are shown in Fig. 7. Two local maxima of the open ocean HDRF are visible for low wind speeds. The first one around the specular point represents the sunglint. It becomes broader and

decreases in intensity as the wind speed increases. Furthermore, with increasing wind speed, the peak of the sunglint is shifted further towards the horizon. This feature was already confirmed by observations from Su et al. (2002), although it is not present in the reflectances simulated by Jackson and Alpers (2010). The second maximum is characterized by an enhanced





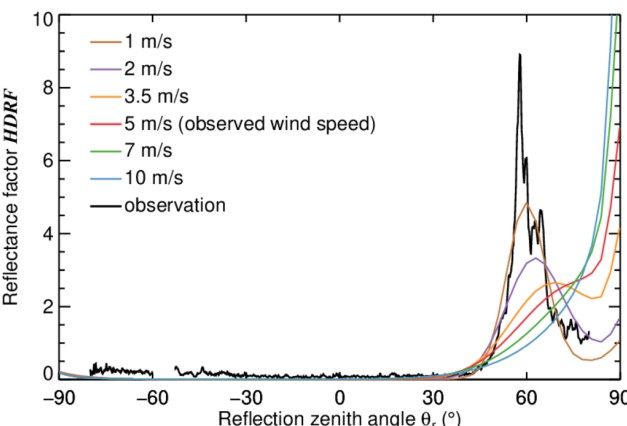

**Figure 7.** Cross-sections of the open ocean HDRF along the solar principle plane simulated with different wind speeds (color-coded) between 1 and $10\,\mathrm{m\,s^{-1}}$ including the observed wind speed of $5\,\mathrm{m\,s^{-1}}$. The black line corresponds to the separated open ocean HDRF.

HDRF towards the horizon which increases with increasing wind speed. This is likely a result of scattering in the atmosphere and multiple scattering at the sea surface. Radiative transfer simulations performed without atmospheric scattering (not shown
here) indicate that the second maximum is reduced or even vanished (for low wind speeds, calm sea surface without multiple scattering). For wind speeds higher than about $3.5\,\mathrm{m\,s^{-1}}$, the second peak becomes dominant while the first peak vanishes in the slope of the second one. The black line in Fig. 7 represents the separated open ocean HDRF. The airborne measurements of the wind speed were extrapolated to $10\,\mathrm{m}$ over ground and amount to about $5\,\mathrm{m\,s^{-1}}$ on average. The comparison between the observation and the simulation for the observed wind speed shows that the maximum HDRF values and their positions do not
coincide. Rather, the separated open ocean HDRF reveals a larger sunglint peak which is located closer to the specular point compared to the simulated open ocean HDRF. The HDRF simulated with a wind speed of $1\,\mathrm{m\,s^{-1}}$ (or even less) compares better to the observations. This indicates that the actual surface roughness of open ocean in the MIZ is significantly reduced because of the wave attenuation between the ice floes, leading to a narrower and more intense sunglint with its maximum closer to the specular point. Nonetheless, the maximum of the separated HDRF is still larger than that of the HDRF simulated with
a wind speed of $1\,\mathrm{m\,s^{-1}}$. The issue that the Cox-Munk simulations underestimate the sunglint intensity was also reported by Su et al. (2002) for large SZAs. However, in contrast to their conclusion, a narrower simulated than observed sunglint was not found in our study, likely due to the variability of surface conditions within the period of observations.

Interestingly, the simulated open ocean HDRF outside the sunglint is significantly lower than the separated one. Nearly independent of the wind speed, the mean simulated HDRF of the shadow side in the solar principle plane ($-90°$ to $0°$) is
around 0.02, which is about one order of magnitude lower than for the separated HDRF. It is likely that these differences are due to horizontal photon transport. In the MIZ some photons reflected from the sea ice surface are scattered in the atmosphere such that they are detected in directions that actually point to open ocean as discussed by Schäfer et al. (2015). This effect may increase the HDRF of open ocean in the MIZ compared to homogeneous ice-free ocean.





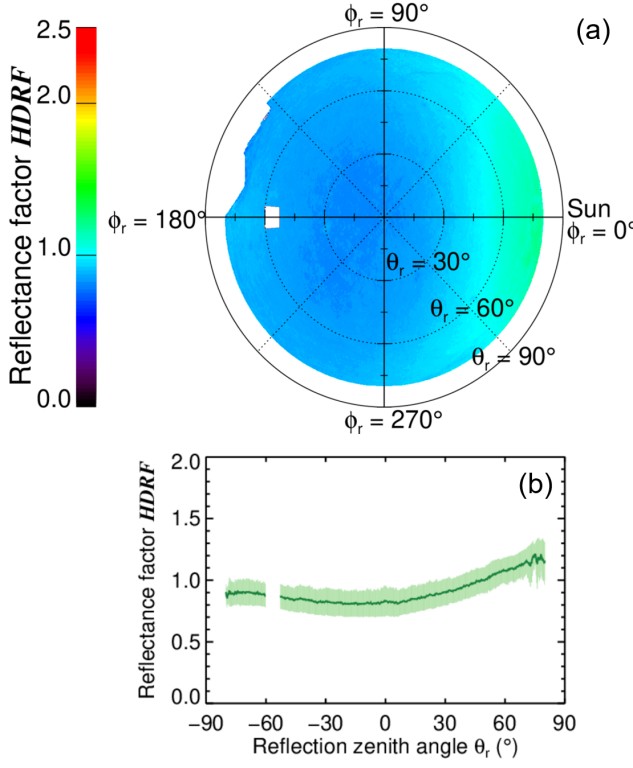

**Figure 8.** Same as Fig.6, but for sea ice. The mean pixel-based sea ice fraction and the mean pixel-based sunglint fraction are missing in (b).

### 4.2 HDRF of snow-covered sea ice

Similar to open ocean (Fig. 6), the average HDRF of the separated snow-covered sea ice areas (separated sea ice HDRF) is shown in Fig. 8. Note, that the ranges of the y-axes in Figs. 6b and 8b are different. In comparison to open ocean, the HDRF values are significantly higher for a large part of the angular domain with values around 0.9. The HDRF is slightly enhanced in the forward direction, which is obvious in both the polar plot (Fig. 8a) and the HDRF along the solar principal plane (Fig. 8b), and is in accordance with the literature (e. g., Bourgeois et al., 2006; Gatebe et al., 2003; Goyens et al., 2018). Furthermore,

the variability of the separated sea ice HDRF (standard deviation below 0.20) is much lower than that of the separated open ocean HDRF due to the high sea ice fraction providing a higher number of images used for averaging.

In order to assess potential differences between the HDRF of homogeneous snow-covered sea ice surfaces and sea ice areas in the inhomogeneous MIZ, the separated sea ice HDRF is compared to homogeneous sea ice and snow HDRFs obtained from two studies (Goyens et al., 2018; Carlsen et al., 2020). The ground-based fish-eye camera measurements described by Goyens

et al. (2018) were performed on landfast sea ice in the southern Baffin Bay in May and June 2015. The HDRF of three different sea ice surface types (bare ice, snow-covered ice and ponded ice) were analyzed. The airborne observations made by Carlsen et al. (2020) were performed over homogeneous snow surfaces on the Antarctic Plateau in December 2013. The SZA during

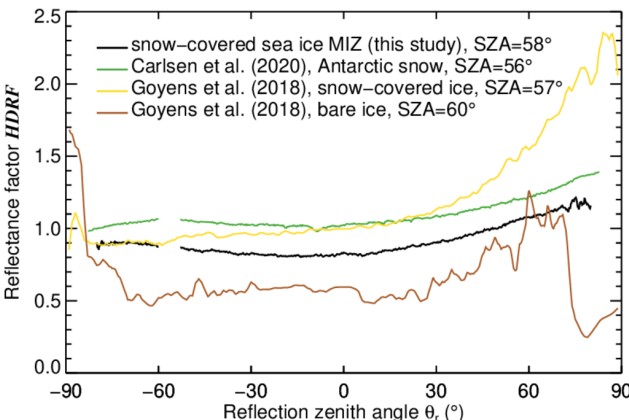

**Figure 9.** Comparison of the cross-sections of the HDRF of snow, snow-covered and bare sea ice along the solar principle plane obtained from different studies (color-coded), see text for details.

these measurements was similar to the SZA observed in our case study (about $58°$). The comparison of the HDRF along the solar principle plane is shown in Fig. 9.

The snow HDRF from Carlsen et al. (2020) (green line) is larger than the separated sea ice HDRF (black line) for all reflection directions of the solar principle plane. Although the shape of both HDRFs is similar, the difference between them (0.19 at nadir) decreases with increasing reflection zenith angle in the forward direction (min. 0.12 at $59°$). The HDRF of snow-covered sea ice (yellow line) observed by Goyens et al. (2018) agrees with the separated sea ice HDRF only for reflection zenith angles less than $-60°$. With increasing reflection zenith angle, the difference between the HDRFs increases to 0.95. While the

minimum of the separated sea ice HDRF seems to be around nadir, the snow-covered sea ice HDRF from Goyens et al. (2018) is lowest in the backward direction (at about $-60°$). However, the well documented increase of the HDRF towards the horizon in forward direction is also visible in their observations. The larger anisotropy of the snow-covered sea ice HDRF observed by Goyens et al. (2018) can be explained by the smaller snow grain size (e. g., Warren et al., 1998). Although the exact optical-equivalent snow grain size is not known, the snow sampled by Goyens et al. (2018) is referred to as *cold snow* which typically

has smaller grains than snow in the MIZ. Analyzing sections with high sea ice fraction, the snow grain size retrieval following the approach by Zege et al. (2011) showed grain sizes of about $200\,\mu\mathrm{m}$ during ACLOUD. However, the optical-equivalent snow grain size on the Antarctic Plateau retrieved by Carlsen et al. (2020) was also significantly smaller ($78\,\mu\mathrm{m}$) than in the MIZ, but the anisotropy of their HDRF is not enhanced compared to the anisotropy of the separated HDRF. The variability of the snow-covered sea ice HDRF observed by Goyens et al. (2018) is slightly larger than that of the other aforementioned

HDRFs. One reason might be the smaller footprint of ground-based measurements compared to airborne observations and, thus, the ability to better resolve small-scale surface roughness of the snow. The HDRF of bare ice (brown line) is significantly lower than the separated sea ice HDRF (0.26 on average), only for a reflection zenith angle of about $60°$ the values are similar. Although the HDRF increases towards the horizon in forward direction, the shape of the bare ice HDRF is very irregular. Its





variability is even larger than that of the snow-covered sea ice HDRF. According to Goyens et al. (2018) this is due to the
presence of thawed ice nearby highly reflective ice grains, which often occurs at the beginning of the melt season.

In summary, the separated sea ice HDRF is lower compared to the HDRFs of homogeneous snow-covered surfaces. Although the snow grain size is larger in the MIZ, the weak absorption of snow for visible wavelengths cannot explain these differences. While the measurements from Carlsen et al. (2020) were performed in the green camera channel (central wavelength of 538 nm), the red channel (628 nm) measurements by Goyens et al. (2018) were used for comparison. However, the
spectral difference of white snow in the visible range should be small. Rather, the HDRF in the MIZ could be slightly smaller due to the effect of horizontal photon transport mentioned above. However, since several other properties (such as snow grain shape, snow pack density, or impurity concentration) can affect the snow HDRF, the comparison illustrates the variability of the snow and sea ice HDRF in Arctic environments.

## 5  HDRF of the MIZ as function of sea ice fraction

In the next step, the average HDRF of the inhomogeneous sea ice–open ocean surface of the MIZ (mean MIZ HDRF, obtained by averaging all images without separation) is analyzed (Fig. 10a). Despite the high sea ice fraction, the mean MIZ HDRF shows features of the HDRFs of both open ocean and sea ice surfaces. The strongly enhanced reflectance in the sunglint region is clearly visible. However, because of the high sea ice fraction, its maximum HDRF (about 2.6) is significantly lower compared to the separated open ocean HDRF (see Fig. 6a). Outside the sunglint but still in forward direction, the slightly enhanced HDRF
characteristic for the sea ice surface is imprinted in the mean MIZ HDRF (compare Fig. 8a). For all other directions, the HDRF is more or less isotropic with values slightly lower (mean of 0.74 on the shadow side) than observed in the sea ice HDRF (0.85), due to the contribution of open ocean surfaces.

In the following, the HDRF of the MIZ is reconstructed ($\mathrm{HDRF_{recon}}$) assuming a linear combination of individual HDRFs of open ocean $\mathrm{HDRF_{ocean}}$ and sea ice $\mathrm{HDRF_{ice}}$ weighted by the sea ice fraction $f_\mathrm{ice}$:

$$\mathrm{HDRF_{recon}}(f_\mathrm{ice}) = f_\mathrm{ice} \cdot \mathrm{HDRF_{ice}} + (1 - f_\mathrm{ice}) \cdot \mathrm{HDRF_{ocean}}(f_\mathrm{ice}). \tag{4}$$

Firstly, it is tested if the mean MIZ HDRF observed in this case study can be reproduced by the reconstructed MIZ HDRF calculated with Eq. 4 and using the observed sea ice fraction and the separated open open and sea ice HDRFs (Figs. 6 and 8). Then, the reconstructed MIZ HDRF is calculated for arbitrary sea ice fractions. To account for the dependence between surface wind speed, sea ice fraction and sunglint in the MIZ, simulations of $\mathrm{HDRF_{ocean}}$ for different $f_\mathrm{ice}$ are used here. These
simulations are performed in the same way as described in Sect. 4.1 except that the input surface wind speed $v_\mathrm{eff}$ is parametrized as a linear function of the sea ice fraction:

$$v_\mathrm{eff} = v_\mathrm{meas} \cdot (1 - f_\mathrm{ice}). \tag{5}$$

$v_\mathrm{eff}$ is considered as an effective wind speed, that would produce the same surface roughness and, thus, the same open ocean HDRF if the ocean was ice-free. $v_\mathrm{meas}$ is the wind speed measured at flight altitude and scaled to 10 m. It has to be noted that





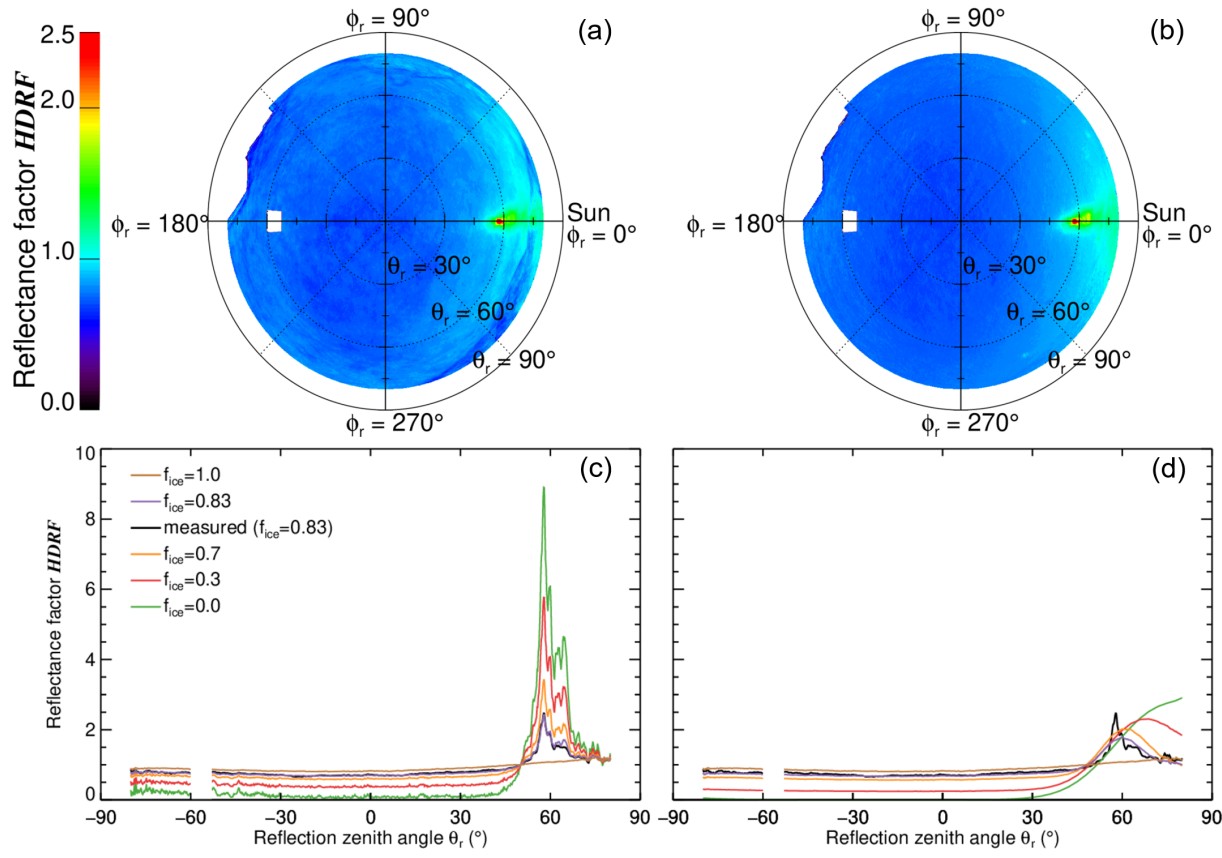

**Figure 10.** Polar plot of (a) the mean MIZ HDRF (obtained by averaging over the entire sequence of images without separation), and (b) the reconstructed MIZ HDRF (calculated from the separated HDRFs of sea ice and open ocean according to Eq. 4 using a sea ice fraction of 0.83). (c) Cross-section of the reconstructed MIZ HDRF along the solar principal plane obtained from the separated HDRFs for different sea ice fractions $f_{ice}$ (color-coded) and the mean MIZ HDRF obtained without separation (black). (d) same as (c), but sea ice fraction-dependent simulations were used for the open ocean HDRF, see text for details.

this very basic relation between surface wind speed and sea ice fraction aims only to illustrate the effects in a qualitative view. Numbers may change if observations for different sea ice conditions are available.

The reconstructed MIZ HDRF calculated for $f_{ice} = 0.83$ (Eq. 4) is shown in Fig. 10b and is compared to the mean MIZ HDRF (Fig. 10a). The difference between both HDRFs is less than 0.1 for more than 96 % of the pixels. For more than 86 % of the pixels, the difference lies within the uncertainty range of the HDRF measurements (6 %). The reconstructed
MIZ HDRF appears more smoothed than the mean MIZ HDRF for statistical reasons. The smoothness was quantified by the standard deviation of the HDRF calculated with respect to all reflection directions of the shadow side (to exclude the sunglint contribution). For the reconstructed MIZ HDRF the standard deviation is slightly lower (0.035, 4.7 % of the mean value) than for the mean MIZ HDRF (0.044, 5.9 % of the mean value). The difference between both HDRFs originates from the calculation





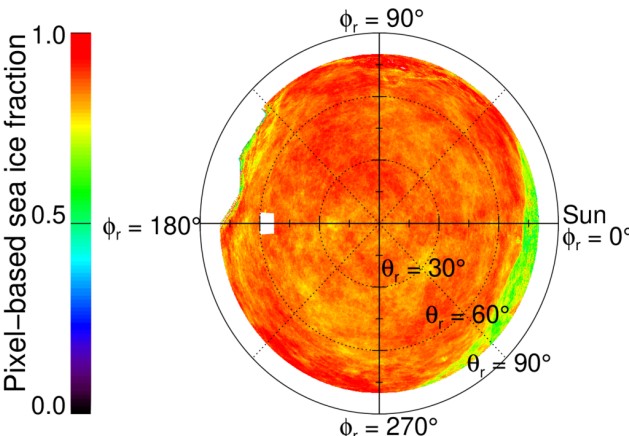

**Figure 11.** Sea ice fraction observed at each pixel throughout the entire image sequence (pixel-based sea ice fraction).

of the reconstructed MIZ HDRF with the mean sea ice fraction, where homogeneous conditions are assumed. This means that
the sea ice fraction is assumed to be uniform for each pixel (direction). In the observations this homogeneity is not given due
to the limited number of images used for averaging. Figure 11 shows a polar plot of the pixel-based sea ice fraction, that
was introduced in Sect. 4. It is obvious, that for each pixel the pixel-based sea ice fraction is different, covering a wide range
between 0.5 and 1.0. Pixels which observed sea ice more often than on average (fraction $> 0.83$), have a larger mean MIZ
HDRF compared to the reconstructed MIZ HDRF and vice versa. This pixel-wise distribution of the sea ice and open ocean
surfaces of the individual images is still imprinted in the mean MIZ HDRF. A higher number of images, which could be reached
by, e. g., increasing the sampling frequency (currently 1/6 Hz), would reduce such effects and the pixel-based sea ice fraction
would become more homogeneous.

Figures 10c and 10d show the cross-section of the reconstructed MIZ HDRF along the solar principal plane for different sea
ice fractions, including the observed mean sea ice fraction of 0.83, and the mean MIZ HDRF (black lines). The open ocean
and sea ice HDRFs are represented by the sea ice fractions of 0 and 1, respectively. While the dependence of the open ocean
HDRF on the sea ice fraction (Eq. 5) is neglected in Fig. 10c, the sea ice fraction-dependent simulations are used in Fig. 10d.
Based on the linear combination (Eq. 4), both panels show a decrease of the magnitude of the sunglint peak with increasing
sea ice fraction. However, in Fig. 10d, also the shift of the sunglint towards the horizon is visible as the effective wind speed
increases with decreasing sea ice fraction. Furthermore, the maximum of the sunglint contribution is significantly decreased
in Fig. 10d. For sea ice fractions lower than observed, this is partly due to the higher effective wind speed. Additionally, the
sharp peak that is visible in in Fig. 10c as a result of the pure specular reflection observed in some images, is not present in the
simulations, which reduces the HDRF maximum. This difference in the sunglint region is the major discrepancy between the
reconstructed MIZ HDRF for the observed sea ice fraction (0.83, purple line in Fig. 10d) and the mean MIZ HDRF. Outside
the sunglint, they show good agreement. For 65 % of the directions of the solar principle plane the relative difference between
both HDRFs is within the uncertainty of the HDRF measurements.





This analysis indicates that the linear construction of the HDRF in the MIZ from individual HDRFs of sea ice and open ocean weighted by the sea ice fraction is possible. However, the impact of the sea ice distribution on the surface roughness and, thus, the open ocean HDRF in the MIZ has to be considered.

## 6  Conclusions

Reflected radiance measurements from an airborne 180° fish-eye camera were used to analyze the surface HDRF in the MIZ north of Svalbard in June 2017. The HDRF was calculated during cloud-free conditions for a 20-minute sequence of 138 camera images covering different sea ice fractions. The sea ice and open ocean surfaces were separated by applying a sea ice mask with different reflectivity and color ratio thresholds.

From the separated images, the averaged HDRFs of open ocean and snow-covered sea ice surfaces in the MIZ were derived.
The number of pixels used for the separated open ocean HDRF was rather low, which led to a higher variability compared to the separated sea ice HDRF. Nevertheless, the separated HDRFs confirmed the general features of the open ocean and sea ice HDRFs reported in literature (e. g.,  Warren et al., 1998; Gatebe et al., 2003; Jackson and Alpers, 2010). However, a comparison of the separated open ocean HDRF with simulations indicated that the common BRDF parametrizations for homogeneous open ocean surfaces as function of the wind speed (Cox and Munk, 1954) partly differ in the MIZ. This is because, firstly, the
wave attenuation between the ice floes in the MIZ (Kohout et al., 2011) leads to a reduced surface roughness compared to a homogeneous open ocean surface with the same surface wind speed. Thus, a narrower and more intense sunglint is present in the MIZ, which may approach pure specular reflection. The irregular-shaped sunglint observed in the separated open ocean HDRF was a result of the highly variable surface roughness associated with the irregular distribution of sea ice and open ocean in the MIZ. Secondly, the reflectance outside the sunglint could be enhanced in the MIZ due to horizontal photon transport
from the sea ice surfaces.

The separated sea ice HDRF was compared with independent measurements from the literature. It was found to be lower than that of homogeneous surfaces. This could be due to horizontal photon transport (Schäfer et al., 2015), complementary to the open ocean HDRF. However, since other unknown properties (such as snow grain shape or surface roughness) affect the snow HDRF, the difference could also originate from the large variety of these properties.

The mean HDRF of the MIZ, calculated as average of all 138 images of the 20-minute section without separation, showed features of the HDRFs of both the sea ice and the open ocean surfaces. Even for rather high sea ice fractions, there is still a contribution from the sunglint in the MIZ which needs to be considered in the analysis of observations. This especially holds regarding the permanent presence of leads in Arctic (e. g.,  Ivanova et al., 2016).

The mean MIZ HDRF was compared to the reconstructed MIZ HDRF, calculated as a linear combination of the separated
HDRFs weighted by the sea ice fraction. The comparison showed good agreement for the measured sea ice fraction with a difference of less than 6 % for 86 % of the pixels. The reconstructed MIZ HDRF appeared more smoothed than the mean MIZ HDRF since the calculation of the reconstructed MIZ HDRF assumes the sea ice fraction to be constant for all reflection directions. In total, this analysis implies that the combination of homogeneous HDRFs is possible. This approach becomes



even more relevant as for most applications only the average sea ice fraction but not the exact distribution of sea ice and open
ocean in the MIZ is known.

However, it was shown that the impact of the ice floe distribution on the surface roughness and, thus, the sunglint in the
MIZ has to be considered for the choice of the open ocean HDRF. To demonstrate these effects, a simple approach was
proposed which parametrized the effective wind speed as a function of the sea ice fraction. The improved results indicated, that
observations or simulations for homogeneous surfaces need to be treated with caution when applied in the MIZ.

This study showed that airborne observations of HDRF, even if limited to a short flight section, can help to characterize
the surface reflection properties of inhomogeneous surfaces in the MIZ. Similar measurements in different environmental
conditions (sea ice fraction, surface wind speed) are needed to fully parametrize the HDRF in such complex scenarios which
likely will dominate the Arctic in a future climate.

*Data availability.* All data measured by the research aircraft *Polar 5* during ACLOUD and used in this study are published on the PANGAEA
database. The radiances measured by the digital camera equipped with fish-eye lens can be found at Jäkel and Ehrlich (2019, https://doi.
org/10.1594/PANGAEA.901024). The spectral irradiance data measured by SMART were published by Jäkel et al. (2019, https://doi.org/
10.1594/PANGAEA.899177). The wind speed was published together with other meterological parameters (Hartmann et al., 2019, https:
//doi.org/10.1594/PANGAEA.902849). The radiosounding used for radiative transfer simulations is available in Maturilli (2017, https://doi.
org/10.1594/PANGAEA.879822) . The HDRF data described by Goyens et al. (2018) can be found on SEANOE (Goyens et al., 2015,
https://doi.org/10.17882/55352).

*Author contributions.* All authors contributed to the discussion of the results and the editing of the article. SB selected the case study,
analyzed the data and drafted the article. SB developed the sea ice mask and performed the radiative transfer simulations. AE initiated the
study. EJ processed the radiance and irradiance data. MW and AE designed the experimental basis of this study.

*Competing interests.* The authors declare that they have no conflict of interest.

*Acknowledgements.* We gratefully acknowledge the funding by the Deutsche Forschungsgemeinschaft (DFG, German Research Foundation)
– Projektnummer 268020496 – TRR 172, within the Transregional Collaborative Research Center "ArctiC Amplification: Climate Relevant
Atmospheric and SurfaCe Processes, and Feedback Mechanisms (AC)[3].





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
