# Peer review of "Airborne measurements of directional reflectivity over the Arctic marginal sea ice zone"

_Atmospheric Measurement Techniques, 2021_

## Author Comment (AC1)

**Answers to comments of Review #1**

We gratefully thank the reviewers for the positive feedback on our submitted manuscript. We appreciate the time they took to extensively read and comment on the given manuscript. The constructive comments are very helpful for the improvement of the manuscript. Our replies to the referees' comments are structured as follows:

*Referee's comments in italic – line numbers according to initially submitted manuscript* Authors' responses in roman – line numbers according to adjusted manuscript. **Citations from the initial and the adjusted manuscript are given in bold.**

In some places the authors seem to discuss findings which are not too surprising and well known. In these places discussion could be slightly shortened and additional literature could be referenced. This is the case in the discussion of the sunglint shape and the Cox and Munk parameterization as well as the impact of "horizontal photon transport".

We considered this valid suggestion and revised some parts of the manuscript. Details to these changes are given in the replies to the specific comments below:

- The shortening of the discussion of the Cox and Munk parametrization is addressed in the answer to comment on line 245.
- The discussion of the horizontal photon transport is shortened drastically, especially for the separated sea ice HDRF. The shortening followed the comment from Referee #2 on lines 310f.

Abstract: Please state somewhere in the abstract that this is a case study for one SZA and a 20minute data set. For some time, I expected more after reading the abstract.

We agree, that our study is very limited to this one 20 min case, which should be addressed already in the abstract. We changed the respective sentence: "Therefore, in this case study, an averaged hemispherical-directional reflectance factor (HDRF) of the inhomogeneous surface (mixture of sea ice and open ocea) in the MIZ is derived using airborne measurements collected with a digital fisheye camera during a 20 minute low-level flight leg in cloud-free conditions." (I. 6–9)

*I.45 – Please name the difference between BRDF and HDRF. BRDF – direct illumination only. HRDF – including diffuse light.*

Yes, this came to short in the original manuscript. We added the following sentence making clear the different nature of the incident radiation (direct/diffuse): "While the BRDF only considers direct illumination from one single direction, also diffuse illumination from the entire hemisphere is taken into account by the HDRF." (I. 46ff.). Later, we give a more detailed introduction to the quantities in Sect. 2.1. For further details we refer in this section to literature (Nicodemus et al., 1977, Schaepman-Strub et al., 2006).

L62 and l.63 – "goniometer" should be "spectrogoniometer" or "goniospectrometer". Otherwise, it is just for measuring angles.

Thanks for your comment. I was not aware of that and changed it accordingly in I. 66 and 67.

**Eq.3 – Why did you omit the "d" in "dFi" e.g.? It is still incremental in i and r, isn'it?**

Since the incident irradiance originates from the entire atmosphere (at least the diffuse component) for the HDRF, *F*i isn't an infinitesimal quantity anymore. The reflected radiances, however, are still infinitesimal quantities, since, in the proper definition of the HDRF, the radiation is reflected into infinitesimal solid angles. However, infinitesimal quantities cannot be measured. Strictly speaking, we derive a hemispherical-conical reflectance factor. However, as the solid angle of the camera pixels are rather narrow, we refer to the quantity derived from the measurements as HDRF, which is commonly done in literature.

To make this more clear, we adjusted the text (l. 102–113):

"Since the illumination under atmospheric conditions is a combination of a direct and a hemispherical diffuse irradiance component with the fractions  $f_{dir}$  and  $f_{diff} = 1 - f_{dir}$ , respectively, both BRDF and BRF cannot be measured practically. Therefore, the hemispherical-directional reflectance factor (HDRF, dimensionless)  $R_{HDRF}$  is introduced (e. g., Schaepman-Strub et al., 2006):

(updated equation 3):  $R_{\text{HDRF}}(\theta_{i}, \varphi_{i}, 2\pi, \theta_{r}, \varphi_{r}) = R_{\text{BRF}}(\theta_{i}, \varphi_{i}, \theta_{r}, \varphi_{r}) \cdot f_{\text{dir}} + R(2\pi, \theta_{r}, \varphi_{r}) \cdot f_{\text{diff}}$

The reflectance factor of the diffuse radiation incident over the entire hemisphere is denoted by  $R(2\pi; \vartheta_r, \varphi_r)$ , were  $2\pi$  refers to the diffuse radiation incidence. The direction of the direct component is given by  $\vartheta_i$  and  $\varphi_i$ . The spectral dependence is omitted here. If the diffuse fraction of the incident radiation is sufficiently small, the HDRF represents a good approximation of the BRF. Since the infinitesimal quantities in Eqs. 1–3 are not measurable, in practice, measurement optics with sufficiently small opening angles are applied to approximate the finite radiances. Thus, from a measurement perspective, the HDRF is obtained by:

(new equation 4):  $R_{\text{HDRF}}(\theta_{i}, \varphi_{i}, 2\pi, \theta_{r}, \varphi_{r}) = \frac{\pi \operatorname{sr} I(\theta_{i}, \varphi_{i}, 2\pi, \theta_{r}, \varphi_{r})}{F(\theta_{i}, \varphi_{i}, 2\pi)}$

To be consistent, we also changed equation 2 and added  $f_{BRDF,id}$  after "constant BRDF of a Lambertian surface" (I. 99):

new equation 2:  $R_{\text{BRF}}(\theta_i, \varphi_i, \theta_r, \varphi_r) = \frac{f_{\text{BRDF}}(\theta_i, \varphi_i, \theta_r, \varphi_r)}{f_{\text{BRDF,id}}} = \pi \operatorname{sr} \cdot f_{\text{BRDF}}(\theta_i, \varphi_i, \theta_r, \varphi_r)$

*I.151 – "inter-calibrated". With which instrument? Obviously SMART, but please state it.*

True. For clarity, we added "with SMART" at the end of the sentence (I. 159f).

*l.*156 - Why does error propagation of two relative errors around 4% for the quotient HDRF not lead to the sum of relative errors 8% as textbooks teach? Please explain your derivation in the manuscript.

We applied the rules of the Gaussian error propagation, which resulted in a total uncertainty of about 6 %. We clarified that by changing the sentence to: "**Thus, using the Gaussian error propagation, the total uncertainty of the calculated HDRF amounts to about 6 %.**" (I. 167f)

I.159 – Please add the original publication "Mayer and Kylling" to the reference here.

Done. (l. 169)

*I.180: Please extend c = Ired/Iblue = 0.95 here.*

We added that (l. 192). Additionally, we defined the variables in one of the previous sentences: "Secondly, a color ratio defined by the ratio of the radiances measured in the red ( $I_{red}$ ) and the blue ( $I_{blue}$ ) camera channel..." (l. 190ff).

Fig. 3 – There is "sunglint" around all ice edges? And "sea ice" around the sunglint? How relevant is this? How large is the error? Please discuss.

The impression that the ice edges are classified as sunglint results only from a poor rendering of the color map while plotting. We fixed this issue and revised Fig. 3b, which now shows much less missclassified pixel around the ice edges. The remaining misclassifications are not avoidable with such a simple sea ice mask.

The misclassification of sunglint as "sea ice" is not easy to remove. Obviously the information content of the three wavelength channels is not sufficient to distinguish the margin of the sun glint area from sea ice. However, the number of pixel affected by this misclassification is still low compared to the entire image. Thus, this misclassification is within the uncertainty range of the sea ice fraction. Since the separated sea ice HDRF does not show any offset in these particular viewing angles, the impact of the missclassification is concluded to be negligible. The effect on the separated open ocean HDRF should be limited to the sunglint margins, where the variability is high anyways. This makes us confident, that the misclassification does not have a significant effect on the following data analysis and interpretation.

We added this discussion in the revised manuscript: "Figure 3b illustrates the surface types identified by the sea ice mask. Together, both panels show the capability of the simple approach to separate between the surface types. Misclassifications mainly affect pixels at the sunglint margin (misclassified as sea ice), which do not have significant implications on the discussion and interpretation in this study. The uncertainty of the sea ice fraction due to the limitations of the sea ice mask is analyzed in the following section. The complete decision process of the sea ice mask is summarized in Fig. 4." (I. 193–199)

*Fig.* 5 – It would be interesting to see variations of this distribution for changes in thresholds as discussed in the text. Think about adding them.

We revised Fig. 5 and added three additional distributions, representing the lowest and the highest sea ice fractions to get an impression of the distribution and the change of the sea ice fractions. However, since the distribution with the lowest sea ice fraction has significantly more misclassifications of sea ice as sunglint, it was not considered in the analysis of the uncertainty range of the mean sea ice fraction. Thus, also the second-lowest distribution is plotted in the new Fig. 5. To describe the new figure, we adapted the text slightly:

"The frequency distribution of the derived sea ice fraction is shown in black in Fig. 5 shows. Images with higher sea ice fractions were more frequent than images dominated by open ocean. The dashed line indicates the mean sea ice fraction of 0.83 sampled during the 20-minutes measurement time interval. The accuracy of the sea ice fraction depends on the choice of the thresholds that are applied in the sea ice mask. In order to estimate the uncertainty related to the choice of the HDRF threshold values, a sensitivity study was performed, slightly varying one of the thresholds while the others were held constant. Two additional distributions, representing the lowest and the highest resulting sea ice fraction, are illustrated in Fig. 5.

The thresholds  $h_1$  and  $h_2$  were varied between the two modes (0.2 to 0.6). Changing  $h_1$  or  $h_2$  by 0.1 leads to a change in sea ice fraction of about 1.2 %.  $h_3$  and the color ratio threshold c were varied between 1.2 and 1.4, and 0.9 and 1.0, respectively. The sensitivity to the sea ice fraction is higher, when  $h_3$  or c are decreased. However, the lower limits of both  $h_3$  and c were chosen such that the amount of obvious misclassification of sea ice as sunglint is limited. As visible in Fig. 5, the averaged sea ice fraction resulting from the variation of the thresholds ranges between 0.79 and 0.86. Thus, the uncertainty of the sea ice fraction due to the sea ice mask is estimated to be less than 4 %." (I. 205–217)

The caption of Fig. 5 is changed to "Frequency distribution of the sea ice fraction resulting from the applied sea ice mask for all images taken within the 20-minute time interval (black) and for adapted sea ice masks (color-coded). The vertical dashed lines represent the resulting mean sea ice fraction."

Updated Fig. 5: Frequency distribution of the sea ice fraction resulting from the applied sea ice mask for all images taken within the 20-minute time interval (black) and for adapted sea ice masks (color-coded). The vertical dashed lines represent the resulting mean sea ice fraction.

**I.215 – A standard deviation of 0.6 for a basis value of 0.11 is really large. Please discuss.**

The standard deviation of 0.6 for a basis value of 0.11 is indeed quite high. We identified misclassifications of sea ice pixels as sunglint (which are assigned to open ocean) as the reason for that, which are not easy to remove with our simple mask. However, the mean value is not heavily affected. We clarified that in the text, also giving the standard deviation if the sunglint criteria (thresholds  $h_3$  and c) would not have been applied. Additionally, we shortened the discussion of the variability due to the low number of open ocean observations since we concluded that this is not the contribution to the variability of the open ocean HDRF. We adjusted the discussion on that: "The standard deviation (blue shading in Fig. 6b) is up to 0.6 outside the sunglint and up to 9.2 in the sunglint region. The high standard deviation outside the sunglint results from misclassifications of sea ice as "sunglint" which are, thus, assigned to open ocean. For each direction, the fraction of open ocean pixels classified as sunglint (calculated as ratio of all images) is referred to as pixelbased sunglint fraction and indicated by the grey line in Fig. 6b. Even on the shadow side, about 10% of the pixels are contributions of misclassified sunglint. Neglecting the sunglint criteria (h₃ and c) would prevent from these misclassifications and reduce the standard deviation to 0.11. Additionally, some of the observed variability results from the high sea ice fraction, which leads to a low number of open ocean pixels in the entire data set. Although the pixel-based sea ice fraction (black line in Fig. 6b), which is derived similarly to the pixel-based sunglint fraction, is higher than **0.8** for most reflection directions, it also reveals significant directional variability." (I. 231–239)

The earlier introduction of the pixel-based sunglint fraction then requires slight changes in the following paragraph: "The irregular shape of the sunglint is likely a result of the low number of observations and is imprinted in the pixel-based sunglint fraction." (I. 242f).

The discussion of the different reasons for the HDRF variability required changes in the comparison of open ocean and sea ice HDRF (Sect. 4.2): "The maximum standard deviation below 0.2 reveals a lower variability compared to open ocean (0.6, including the misclassifications), which indicates that the mean sea ice HRDF is less affected by misclassified pixels." (I. 291ff).

**Fig.6 – What about white lines in the dark part of the image?**

You mean the lines at the top of the image? These are gaps, where open ocean observations were not available within our dataset. They are close to the edge of the image, where, depending on the heading/attitude of the aircraft, not every image delivers data. By changing the color code of the figure, these gaps are now less visible.

Fig.6 – The figure b is quite confusing at a first glance (while it is understandable reading the related paragraphs in the main text). Please add some more details to the caption. Questions that pop up without it: How can sea ice fraction be a function of reflection angle? How can sunglint fraction and sea ice fraction add up to more than 1?

You are right. That could be confusing. We tried to solve this problem by rephrasing the respective sentence "The black line denotes the mean pixel-based sea ice fraction (the portion of images, where the respective pixel of the solar principle plane is assigned to sea ice). The grey line denotes the mean pixel-based sunglint fraction (the ratio of the number of images, where the respective

pixel of the solar principle plane is classified as sunglint to the number of images, where this pixel is assigned to open ocean)." Hopefully, that's more clear now.

*I.245 and I.249 - "already ... 2002" does sound awkward here, as it is obviously correctly considered in the Cox and Munk publication which was based on observations as well. The impact of "reflected skylight" and the need to remove it for their parameterization is discussed there. Please mention.*

We agree that "already 2002" sounds odd. However, I wonder why the simulations of Su et al. (2002) didn't capture the sunglint shift, although they used the Cox & Munk slope distribution. Anyway, also regarding your general command, we reduced the discussion about the open ocean HDRF theory by rephrasing the respective paragraph:

"Cross-sections of the simulated open ocean HDRF for several wind speeds are shown in Fig. 7. For low wind speeds, two local maxima of the open ocean HDRF are visible, representing the sunglint (around the specular point) and the reflection of the diffuse incident radiation towards the horizon. With increasing wind speed, the sunglint distribution becomes broader, while its maximum decreases in intensity and is shifted further to the horizon (e. g., Su et al., 2002). The HDRF peak at the horizon increases with increasing wind speed, which is likely due to an increase of diffuse incident radiation caused by multiple scattering between sea surface and atmosphere. For wind speeds higher than about  $3.5 \text{ m s}^{-1}$ , the diffuse reflection peak becomes dominant while the sunglint vanishes in its slope. The impact of the diffuse radiation (reflected skylight) on the BRDF and a method to remove this offset were discussed by Cox and Munk (1954)." (I. 261–268)

**L256 – Maybe 0.5 m/s - if you could have simulated it - would have provided an even better match.**

Sure. We think so, too. However, simulations with lower wind speed were not supported by libRadtran. That's what we also meant with "**(or even less)**". Of course, the wind speed which would fit best to our observations, remains unclear. We added the sentence "**Unfortunately, simulations** with lower wind speeds were not supported by libRadtran." (I. 273f) to further clarify this.

*I.261 – Do you have "large SZA"? 60 deg doesn't sound too large. Apart from that I have the impression that your results with all their additional uncertainties regarding mix of ice fractions, limited accuracy of glint identification, and true vs effective wind speeds are not suited to analyze the limitations of a theoretical parameterization (Cox and Munk). You could shorten this a bit.*

That's true, although we didn't have the intention to evaluate the parametrization of Cox & Munk. Rather, we wanted to discuss the discrepancy of our measured HDRF compared to the simulated one. You are correct that our limited data set could be responsible for the deviations. However, the SZA of 60° present in our observations is relatively large compared to the observations used by Cox & Munk (their SZA was max. 35°). We wanted to note, that similar discrepancies were observed earlier, without claiming the total correctness of one of the studies. We tried to clarify that by adjusting our wording: "The remaining discrepancy in maximum HDRF between observation and simulation (1 ms-1) could be due to the limited data set (e. g., the variability of the sunglint shape) or the still too high wind speed. However, since similar differences for high SZAs have been observed by Su et al. (2002), this might also be an effect of the larger SZA compared to the original measurements by Cox and Munk (1954)." (I. 276–279) *I.285/286 – This seems like a quasi-constant offset because of the nearby dark open ocean influence. Please state this here.*

It was remarked by Referee #2 that, given an open ocean fraction of 17 % and a change of the separated water HDRF of 0.09 due to possible 3D effects, the opposite effects on the 83 % sea ice cannot explain an offset of 0.1 to 0.2. Thus, these effects should only be minor for sea ice. Instead, we argue that the our HDRF (partly snow-covered ice floes) lies between the HDRF of fully snow-covered surfaces and bare ice (which fits with Fig. 9). We revised the last two paragraphs of the section completely (see below). We refer to the reply on the comment by Referee #2 lines 310f.

L296 – "However, the optical ..." - Why "however"? I was confused first. I was expecting a snow grain size for Goyens, but got one for Carlsen. Please check wording and improve if possible.

Since the argumentation with snow grain size is misleading given that our observed ice surfaces are only partly snow-covered, we revised the last two paragraphs of the section completely (see below).

*I.311 – "horizontal photon transport mentioned above" – I agree, but why not putting this into clearer words. "the nearby dark ocean surface", "the reduced diffuse light due to the nearby dark ocean" …*

As stated above, we spend less attention to the impact of the nearby ocean surface in our revised text. However, one remark on that is still included (see below).

**Revised paragraphs:**

[revised manuscript text omitted]

---

## Author Comment (AC2)

**Answers to comments of Review #2**

We gratefully thank the reviewers for the positive feedback on our submitted manuscript. We appreciate the time they took to extensively read and comment on the given manuscript. The constructive comments are very helpful for the improvement of the manuscript Our replies to the referees' comments are structured as follows:
*Referee's comments in italic – line numbers according to initially submitted manuscript*
Authors' responses in roman – line numbers according to adjusted manuscript. **Citations from the initial and the adjusted manuscript are given in bold.**

*Regarding your figures: please avoid the usage of rainbow colormaps, as they are misleading the perception (see e.g. Borland and Taylor 2007: DOI 10.1109/MCG.2007.323435).*

We changed that

*The study focuses mainly on the ability to mix and unmix the HDRF of two very different surface types. After reading your title I expected a study that particularly concentrates on the MIZ, and thus, on its distinct features and properties and their impact on reflectivity. You discuss only sea ice fraction and note its effect on horizontal photon transport and wave attenuation. Properties of the MIZ like a distinct floe size distribution, the effect of ice edges etc. remain underexplored or are even actively neglected. Improving this could substantially extent the scientific outcome regarding the MIZ.*

We will give an answer to this at the specific comments (esp. comment on line 174)

**Major issues of the present manuscript are:**

*There is a fundamental mismatch between your definition of the MIZ (ice fraction between 15% and 80%) and the ice fraction in your study area (83%) which actually contradicts your claim to have measured in the MIZ. That needs to be discussed in the manuscript.*

As shown in the satellite image (Fig. 1a), the observations origin from an area very close to the sea ice edge. Although our observations exceed the upper limit of the commonly used thresholds of sea ice fraction (Strong and Rigor, 2013) slightly, the characteristics of the observed sea ice which is dominated by separated and irregularly distributed ice floes is closer to the MIZ than to the Central Arctic where open water mostly only occurs in elongated open leads. Thus, we don't see a fundamental mismatch here. Furthermore, Strong and Rigor (2013) defined the thresholds based on satellite observations with limited spatial resolution. Small scale ice floes for example might be undetected by satellite observations with spatial resolutions lower than these ice floes. In addition, the measured HDRF is clearly affected by both surface types (open water and sea ice), especially the sunglint feature is prominent in the mean or constructed HDRF of the MIZ. Thus, we think that, for our and similar radiative applications with high spatial resolution, the strict definition by Strong & Rigor can be relaxed, such that the studied region can be considered as part of the MIZ although it is at the upper threshold of their MIZ definition. In the text, we discuss this by adding:

**"Although the derived sea ice fraction slightly exceeds the upper limit of the MIZ definition given by Strong and Rigor (2013), the observations origin from an area very close to the sea ice edge and are characterized by separated and irregularly distributed ice floes typical for the MIZ."** (l. 218ff)

*Following your method description, you use the HDRF first as a threshold to separate surface classes, create your sea ice mask and derive a coverage fraction. Subsequently, you use the averaged HDRF of both classes and their fraction to derive a reconstructed MIZ HDRF, similar to a weighted mean. You then compare that to the arithmetic mean.*

- o *First thought: if the ice fraction would be homogeneous in all directions, the only difference between reconstructed MIZ HDRF and mean MIZ HDRF would be some reclassified pixels in the area of sunglint or ice, slightly changing the weights. Apart from that, the average would be mathematically identical and thus rather insufficient for comparison. So your discussion of errors induced by differences in ice fraction across all directions is correct, but most probably that will remain the only source of error between reconstructed and mean MIZ HDRF even in larger data sets.*
- o *Second thought: Using the HDRF to classify pixels/directions still seems reasonable, given that you only have this one optical data set. Which, in addition, is also very limited in size. Nevertheless, it would be important for the validity of the method to at least separate training and test data, e.g., by splitting your flight into subsets. You can still give the overall mean from all images as a result, but the evaluation needs separate datasets for training and testing.*
- o *This needs to be addressed and discussed in the text. Otherwise, this dual use and mathematical dependency makes your results vulnerable, since a comparison to independent data is lacking so far.*

Yes, if the data set is sufficiently large, the sea ice fraction becomes statistically more robust and would be the same for each pixel. The only difference between mean and constructed HDRF would result the excluded pixels.

We agree. To check, whether this construction of the MIZ HDRF from individual HDRFs is possible (given that all environmental conditions are considered correctly), it's not advantageous to use the same data set, since differences here only occur due to the mathematics, you just discussed. Therefore, we apply your suggestion and randomly split our data set. We separate one subset (separation subset) and compare it to the second subset (test dataset, unseparated). Within our short leg, the exact handling of the environmental conditions is ensured. However, for the analysis of the individual HDRFs (Sect. 4), the entire data set is used.

To include the description and procession of the subsets (and due to further comments), we revised Sect. 5 (see below comment on lines 361-365).

*Title: Fine, but at the latest in the abstract, the limited data basis of a 20min flight at only one location should be addressed.*

*Abstract: Please note that your study is based on only one flight, otherwise readers expect a greater data basis (as mentioned). The abstract also lacks concrete results like the deviations in the open ocean that you found. The last sentence should not be stated as such, since you have not accounted for all special conditions of the MIZ in your study (as previously mentioned).*

We agree, that our study is very limited to this one 20 min case, which should be addressed already in the abstract. We changed the respective sentence: "**Therefore, in this case study, an averaged hemispherical-directional reflectance factor (HDRF) of the inhomogeneous surface (mixture of sea ice and open ocean) in the MIZ is derived using airborne measurements collected with a digital fish-eye camera during a single 20 minute low-level flight leg in cloud-free conditions.**" (l. 6–9)

Additionally, we changed the last three sentences: "**It is shown that the open ocean HDRF in the MIZ differs from homogeneous ocean surfaces due to wave attenuation. Using individual HDRFs of both surface types and the sea ice fraction, the mixed HDRF describing the directional reflectivity of the inhomogeneous surface of the MIZ was retrieved by a linear weighting procedure. Accounting for the wave attenuation, good agreement between the average measured HDRF and the constructed HDRF of the MIZ was found for the presented case study.**" (l. 11–15)

*19: sea ice–albedo feedback (long dash "en dash")*

We changed it accordingly (l. 20).

*28f: "mixture of sea ice and open ocean, e.g., [..] melt ponds." Needs reformulation*

Indeed, melt ponds do not belong to open ocean. However, they affect the HDRF in the MIZ. We reformulated the sentence: "**However, large areas of the Arctic ocean are characterized by a mixture of sea ice and open ocean. The open water might be formed by leads or polynyas, while sea ice may also be covered by melt ponds (e. g., Hoffman et al., 2019).**" (l. 28f)

*37: "is dominated by freshly.."*

We changed it accordingly (l. 37).

*37: Why therefore? Do you refer here to sentence 36f?*

Yes, we intended to refer to the sentence before the previous one (concerning the summertime MIZ). We made that clear by adjusting the sentence: "**The summertime MIZ widening trend highlights the necessity for characterizing the radiative properties of the mixture of sea ice and open ocean, which is needed to better quantify the complex radiative processes in such areas.**" (l. 38ff)

*40f: That statement needs a reference.*

This simplification is based on the definition of the broadband solar energy budget ($F_{\mathrm{net}} = F^\downarrow - F^\uparrow$). The upward irradiance $F^\uparrow = \alpha \cdot F^\downarrow$ can be written as the fraction of the downward irradiance $F^\downarrow$ that is reflected at the surface due to the (broadband, hemispherical) albedo $\alpha$. We think, that this is just a mathematical relation that doesn't need necessarily a reference. However, maybe that's not

obvious. Thus, we added a reference introducing this simplification in more detail: "**(e. g., Stapf et al., 2020)**" (l. 42).

*61: the camera does not provide "instantaneous reflectance observations". Only in combination with a SMART or similar you can derive reflectance.*

That's true. The formulation is not very accurate. From the camera, only the radiances can be derived. Accordingly, we changed "**reflectance**" to "**radiance**" (l. 63). After this sentence we additionally added the sentence "**In both cases, the downward spectral irradiance measured by a spectral hemispherical radiometer is required for the derivation of the HDRF.**"

*78ff: High resolution could be added as a motivation for the camera here*

To motivate the camera with both the hemispherical coverage and high spatial resolution, we modified the sentence: "**In this study, the hemispherical coverage and high spatial resolution of airborne fish-eye camera observations are used to characterize the HDRF of a mixture of open ocean and snow-covered sea ice surfaces in the MIZ.**" (l. 81f)

*81ff: It is a matter of taste, but it has become more common to formulate the scientific questions at the end of your introduction and not to give an extended table of contents, since the overall structure of your manuscript is clear anyways.*

We tried to include your suggestion by including the scientific questions and giving an outlook how to answer them in the course of the study. We changed the respective passage: "**After the introduction of the used quantities, as well as the observations and instruments in Section 2, the following two questions will be discussed: (i) How does the HDRF of the individual contributions of open ocean and sea ice in the MIZ differ from homogeneous surfaces? To answer this, the contributions of the individual surface types on the observed mixed scenes are separated applying a sea ice mask (Section 3) and compared to HDRFs of homogeneous surfaces (Section 4). (ii) Is it possible to combine the HDRF of the individual surfaces, weighted by the sea ice fraction, to obtain a representative MIZ HDRF? For this, the data set is divided into two subsets. The previously separated HDRFs of one subset are recombined and compared to the (unseparated) mean of the other subset (Section 5).**" (l. 82–89)

*101: why do you use different notations HDRF and $R_{HDRF}$?*

You are right, that's not very consistent. The intention to use the variable name HDRF later on was to avoid lengthy subscripts. In contrast, a physically more proper notation was chosen for the definition of the quantities, also to indicate the difference between the BRDF (reflectance distribution function, variable $f$) and the different reflectance factors $R$. Based on your comment, we decided to replace the variable HDRF below by $R_{HDRF}$, followed by another subscript. This only affects Eq. 6 (former Eq. 5) together with the explanations of the variable names in the sentence before (l. 331) as well as Fig. 4.

*101: was introduced*

We don't intend to change the tense here. Except the mentioning of the HDRF in the introduction, we introduce this quantity to the reader right here. In our opinion, there is no need to state the sentence in past tense.

*101f: why however?*

The word "however" can be omitted.

Rereading the paragraph concerned by the previous two comments and concerning the comment of Referee #1 on Eq. 3, we thought of reformulating it a bit (l. 102–113):

"**Since the illumination under atmospheric conditions is a combination of a direct and a hemispherical diffuse irradiance component with the fractions $f_{dir}$ and $f_{diff} = 1 - f_{dir}$, respectively, both BRDF and BRF cannot be measured practically. Therefore, the hemispherical-directional reflectance factor (HDRF, dimensionless) $R_{HDRF}$ is introduced (e. g., Schaepman-Strub et al., 2006):**

**(updated equation 3):** $R_{\mathrm{HDRF}}(\theta_{\mathrm{i}}, \varphi_{\mathrm{i}}, 2\pi, \theta_{\mathrm{r}}, \varphi_{\mathrm{r}}) = R_{\mathrm{BRF}}(\theta_{\mathrm{i}}, \varphi_{\mathrm{i}}, \theta_{\mathrm{r}}, \varphi_{\mathrm{r}}) \cdot f_{\mathrm{dir}} + R(2\pi, \theta_{\mathrm{r}}, \varphi_{\mathrm{r}}) \cdot f_{\mathrm{diff}}$

**The reflectance factor of the diffuse radiation incident over the entire hemisphere is denoted by R($2\pi;\vartheta_{\mathrm{r}},\varphi_{\mathrm{r}}$), were $2\pi$ refers to the diffuse radiation incidence. The direction of the direct component is given by $\vartheta_{\mathrm{i}}$ and $\varphi_{\mathrm{i}}$. The spectral dependence is omitted here. If the diffuse fraction of the incident radiation is sufficiently small, the HDRF represents a good approximation of the BRF. Since the infinitesimal quantities in Eqs. 1–3 are not measurable, in practice, measurement optics with sufficiently small opening angles are applied to approximate the finite radiances. Thus, from a measurement perspective, the HDRF is obtained by:**

**(new equation 4):** $R_{\mathrm{HDRF}}(\theta_{\mathrm{i}}, \varphi_{\mathrm{i}}, 2\pi, \theta_{\mathrm{r}}, \varphi_{\mathrm{r}}) = \dfrac{\pi\,\mathrm{sr} \cdot I(\theta_{\mathrm{i}},\varphi_{\mathrm{i}},2\pi,\theta_{\mathrm{r}},\varphi_{\mathrm{r}})}{F(\theta_{\mathrm{i}},\varphi_{\mathrm{i}},2\pi)}$

To be consistent, we also changed equation 2 and added $f_{\mathrm{BRDF,id}}$ after "**constant BRDF of a Lambertian surface**" (l. 99):
new equation 2: $R_{\mathrm{BRF}}(\theta_{\mathrm{i}}, \varphi_{\mathrm{i}}, \theta_{\mathrm{r}}, \varphi_{\mathrm{r}}) = \dfrac{f_{\mathrm{BRDF}}(\theta_{\mathrm{i}},\varphi_{\mathrm{i}},\theta_{\mathrm{r}},\varphi_{\mathrm{r}})}{f_{\mathrm{BRDF,id}}} = \pi\,\mathrm{sr} \cdot f_{\mathrm{BRDF}}(\theta_{\mathrm{i}}, \varphi_{\mathrm{i}}, \theta_{\mathrm{r}}, \varphi_{\mathrm{r}})$

*111: I would recommend omitting "(Norway)" here, since Svalbard is a known geolocation and the political affiliation to Norway is regulated in a somewhat more complex manner*

Ok. We agree and deleted it.

*111: 19 flights each makes me wonder why you only used one? That needs to be included here.*

We mentioned that several lines below, that we chose this specific flight since we intended to perform a case study in cloud-free conditions (present during that day, 25 June). Maybe, the term "case study" is important. We slightly changed the respective sentence: "**The data analyzed for the present case study were obtained during the flight on 25 June 2017 (flight number 23) performed**

**under cloudless conditions.**" (l. 123). In the remainder of the paragraph we describe the choice of the 20 minute interval which should ensure stable environmental conditions.

*Figure 1: "dots point" -> "dots indicate/show/…"*

Yes, you are right that this is confusing. We changed it to "**The blue and green dots indicate…**".

*116: "cloudless" -> "clear sky"*

We believe that the term "clear sky" would refer to conditions with neither clouds nor aerosols in the pathway of the incident solar radiation. Thus, we prefer the term "cloudless", indicating that the atmosphere is free of clouds, but other atmospheric constituents can be present.

*131: "served as reference in this study."*

That's a good suggestion and makes it more clear. Accordingly, we added "**… in this study**" at the end of the sentence (l. 139)

*148: It is great that you give error estimates for each single component of your workflow, but partially, it remains unclear how you retrieve them exactly. Like 4.2% here.*

True. We refer to the uncertainties from Carlsen et al. (2020), who used the same fish-eye camera with similar characteristics. The uncertainties stem from sensor characteristics (0.5 %), the radiometric calibration (4 %) and the geometrical calibration and attitude correction (combined 1 %). The geometric calibration was performed differently, however, the estimated uncertainty is considered to be similar. Although Carlsen et al. (2020) stated a total uncertainty of 4.5 %, the Gaussian propagation of the three uncertainties mentioned here leads to a total uncertainty of 4.2 %. We changed the sentence to "**The uncertainty of the radiometric calibration of the fish-eye camera was estimated by 4 %, further uncertainties stem from the sensor characteristics and the combination of geometric calibration and aircraft attitude correction (0.5 % and 1 %, resp., Carlsen et al., 2020), leading to a total uncertainty of the fish-eye camera radiance measurements of about 4.2 %.**" (l. 154ff). The SMART uncertainties are given in Bierwirth et al. (2009), everything deviating from their uncertainty analysis is given in the text, leading to the given updated uncertainty.

*151: What do you mean by "the camera was inter-calibrated"?*

We added "**with SMART**" at the end of the sentence (l. 159f). Hopefully, that clarifies that we compared the camera measurements (pixels around nadir only) to radiance measurements available from SMART in the spectral range of the camera channels. Then, the camera radiances were adjusted according to the correlation. This comparison is discussed in detail by Ehrlich et al. (2019).

*153f: Would be good to briefly put the red channel in relation to the spectrum here and discuss what impact you expect on the results. Inaccuracies in other channels are not a good argument to use the red channel, the data set could simply be inappropriate.*

We did that by replacing the respective passage by "**The radiances measured in the red channel showed the best agreement in the instrument intercomparison (root mean square deviation of 0.01 W m$^{-2}$ nm$^{-1}$ sr$^{-1}$ and the correlation coefficient of 0.98 between SMART and camera radiances, Ehrlich et al., 2019). The choice of the channel has only minor impact on the results presented here. While the reflectance of snow is spectrally neutral in the visible range, the spectral open ocean albedo is only slightly higher in the blue than in the red channel. Therefore, it was decided to use the red camera channel in the following.**" (l. 160–164)

*168: On which basis did you select your 138 images from 200(?) captured during 20min.? This must be added here, otherwise all distribution functions are difficult to evaluate, since it must be assumed that they originate from a subjective subset.*

We had to exclude images from turns with roll or pitch angles larger than 5°. That's why not all images did it into the analysis. The excluded images mostly originate from a longer turn separating two horizontal legs. 138 images remained. We changed the sentence to "**The set of images of a low-level flight section of 20 min was filtered for larger turns with roll or pitch angles larger than 5°. The remaining 138 fish-eye camera images were analyzed.**" (l. 179f)

*174f: Why do you exclude data between both modes, most likely from "ice floe edges". Isn't that a major feature of the MIZ? Can you detect any impact of the floe size distribution in your data? 3% doesn't sound much for this dataset, but what about others with a higher fraction?*

Fig. 3b shows that the exclusion mostly affects the ice floe edges or bright meltponds. However, it's also visible, that the fraction of excluded pixels is higher in areas with high sea ice fraction, but a large number of floes. However, to characterize the impact of the floe edges was beyond the scope of this study. Rather, our aim was to extract the HDRF of snow-covered sea ice and open ocean surface. To obtain this, we think it is safer to exclude the ice floe edges as they do not belong to either of them. The amount of excluded data is within the sea ice fraction uncertainty.

The impact of the ice floe edges should be imprinted in the comparison of the mean measured HDRF with the constructed HDRF (Fig. 10). While the ice floe edges are included in the mean MIZ HDRF, the reconstructed HDRF is based on the separated HDRF excluding the edges. However, most of the differences likely are caused by the non-uniform sea ice fraction.

The impact of ice floe distribution (also regarding the wave development between the floes, which is important for the open ocean HDRF) is an interesting problem. We will think about mentioning that in the outlook as a motivation for upcoming studies, please have a look at the last paragraph of the revised conclusions.

*180: add the ratio here c=red/blue*

We added that (l. 192). Additionally, we defined the variables in one of the previous sentences: "**Secondly, a color ratio defined by the ratio of the radiances measured in the red ($I_{red}$) and the blue ($I_{blue}$) camera channel…**" (l. 190ff).

*183: You state "the mask is capable to separate…". When I look at Fig 3, it doesn't seem so: A significant portion of the sunglint area is masked as ice, all ice edges that you claimed to omit are marked as sunglint. That needs to be better discussed.*

The impression that the ice edges are classified as sunglint results only from a poor rendering of the color map while plotting. We fixed this issue and revised Fig. 3b, which now shows much less missclassified pixel around the ice edges. The remaining misclassifications are not avoidable with such a simple sea ice mask.

The misclassification of sunglint as "sea ice" is not easy to remove. Obviously the information content of the three wavelength channels is not sufficient to distinguish the margin of the sun glint area from sea ice. However, the number of pixel affected by this misclassification is still low compared to the entire image. Thus, this misclassification is within the uncertainty range of the sea ice fraction. Since the separated sea ice HDRF does not show any offset in these particular viewing angles, the impact of the missclassification is concluded to be negligible. The effect on the separated open ocean HDRF should be limited to the sunglint margins, where the variability is high anyways. This makes us confident, that the misclassification does not have a significant effect on the following data analysis and interpretation.

We added this discussion in the revised manuscript: "**Figure 3b illustrates the surface types identified by the sea ice mask. Together, both panels show the capability of the simple approach to separate between the surface types. Misclassifications mainly affect pixels at the sunglint margin (misclassified as sea ice), which do not have significant implications on the discussion and interpretation in this study. The uncertainty of the sea ice fraction due to the limitations of the sea ice mask is analyzed in the following section. The complete decision process of the sea ice mask is summarized in Fig. 4.**" (l. 193–199)

*187f: "The sea ice fraction refers to the portion of the total horizontal surface that is covered by ice."*

We changed the sentence accordingly (l. 201f).

*Figure 5: What do you mean here by "all images". Sounds like you did not only use your 138 images but more. Why?*

The 138 images are all images within the leg we chose (see above). So, we used all 138 images.

*200ff: This sentence is not self-explanatory. What do you mean by extreme cases, how do you derive an uncertainty of 4%?*

We agree, that the sentence is confusing for anyone who is not familiar to our study. We slightly revised this section to make our sensitivity study more clear. Based on the comment on Fig. 5 of Referee #1, we also included additional distributions to Fig.5, leading to a slight change in the text.

"**The frequency distribution of the derived sea ice fraction is shown in black in Fig. 5 shows. Images with higher sea ice fractions were more frequent than images dominated by open ocean. The dashed line indicates the mean sea ice fraction of 0.83 sampled during the 20-minutes measurement time interval. The accuracy of the sea ice fraction depends on the choice of the thresholds that are applied in the sea ice mask. In order to estimate the uncertainty related to the choice of the HDRF threshold values, a sensitivity study was performed, slightly varying one of the thresholds while the others were held constant. Two additional distributions, representing the lowest and the highest resulting sea ice fraction, are illustrated in Fig. 5.**

**The thresholds $h_1$ and $h_2$ were varied between the two modes (0.2 to 0.6). Changing $h_1$ or $h_2$ by 0.1 leads to a change in sea ice fraction of about 1.2 %. $h_3$ and the color ratio threshold $c$ were varied between 1.2 and 1.4, and 0.9 and 1.0, respectively. The sensitivity to the sea ice fraction is higher, when $h_3$ or $c$ are decreased. However, the lower limits of both $h_3$ and $c$ were chosen such that the amount of obvious misclassification of sea ice as sunglint is limited. As visible in Fig. 5, the averaged sea ice fraction resulting from the variation of the thresholds ranges between 0.79 and 0.86. Thus, the uncertainty of the sea ice fraction due to the sea ice mask is estimated to be less than 4 %.**" (l. 205–217)

*215f: I think there are several physical explanations and not only a statistical reason. What about wave induced heterogeneity in the sunglint area resolved by the high-resolution images or ice edges classified as sunglint in all different directions?*

A little bit later in the text we discussed the variability of the sunglint among the images, finding very different shapes and extensions of the sunglint. In this regard, we also mention the dependence of the sunglint extension on the surface roughness, induced by waves and the different wave development between the ice floes. This touches the topic of dependence of MIZ HDRF on ice floe distribution without examining that further.

But, you are right. Wave-induced heterogeneity could also be a reason for the variable sunglint. We also agree, that pixels accidentally classified as sun glint could be responsible for the variability.

However, if these effects were uniform throughout the entire hemisphere, the variability would be significantly reduced (maybe the HDRF outside the sunglint would be slightly enhanced due to the misclassifications). Nevertheless, in the end, this is also a result of the limited statistics. Otherwise, the variability would average out.

*Figure 7: It would be helpful to briefly mention the SZA again in the caption.*

Ok. We added "**The mean observed SZA also used for the simulation was 57.8°.**" at the end of the caption.

*256f: Why didn't you include smaller wind speeds to eventually match your observations? Like 0.3, 0.5, …?*

As mentioned in the text, radiative transfer simulations including smaller wind speeds were not possible with libRadtran. We also state in the text "**(or even less)**" and added "**Unfortunately, simulations with lower wind speeds were not supported by libRadtran.**" (l. 273f), that we think, that even lower wind speeds would match better the observations.

*265ff: What about horizontal photon transport from ice edges below the water surface? Or could it be also caused by "blooming" effects on the CMOS sensor?*

Sure, the photons can also be transported from the sea ice floes below the water surfaces. However, for our analysis, the effect is the same, regardless whether the photons are transported above or below the water surface. In both cases, they are lost from the directions pointing to sea ice, which reduces the sea ice HDRF. If the photons then escape from the open water surface, they enhance the HDRF similar to photons transported above the surface and detected in directions pointing to open ocean.
We checked the raw data for "blooming" effects. They don't occur in the data we use for our analysis, all pixel counts were within the dynamical range of the sensor.

*Figure 8: The wording with "missing here" is misleading, rather use something like "but without the …" or similar.*

That's true. We changed the caption to "**Same as Fig 6, but for sea ice and without the mean pixel-based sea ice fraction and the mean pixel-based sunglint fraction in (b).**".

*310f: If you do a highly simplified budget here, photon loss doesn't seem to have an impact: the open ocean is 0.09 too bright on an area fraction of 17% in your comparison. That equals a darkening of the ice by less than 0.02. Your following sentence referring to the variability is much more important. Given the time of the year I would expect some bare ice surfaces (which I think are also visible in your example images) and thus, an HDRF between snow covered ice and bare ice which you perfectly match.*

That's a very wise comment. We didn't think about this so far. You are absolutely right. The horizontal photon transport needs to be considered in absolute numbers, weighted by the area fraction. Since the sea ice fraction is much larger than that of open ocean, the impact of this effect must be larger over open ocean. The deviation between our HDRF and Carlsen's is larger than 0.02 (0.12-0.19). This means, that the difference must be due to other factors. This changes the view on our discussion. Accordingly, we changed the last to paragraphs of this section (see below). Additionally, we changed the simulated open ocean HDRF mean: "**Nearly independent of the wind speed, the mean simulated HDRF of the shadow side (90° to 270° reflection azimuth angle) is around 0.03, which is about 0.08 lower than for the separated HDRF.**" (l. 280ff) to be able to better compare it to our mean open ocean HDRF (0.11), which is also calculated not only for the solar principle plane.

Revised paragraphs:

"**Compared to the separated sea ice HDRF (black line), the snow HDRF from Carlsen et al. (2020) has a similar shape, but shows a higher magnitude for all reflection directions of the solar principle plane (0.19 at nadir). Likewise, the HDRF of snow-covered sea ice (yellow line) observed by Goyens et al. (2018) is larger than the separated sea ice HDRF (0.16 at nadir), except for reflection zenith angles less than −60°. However, comparing both snow HDRFs, significant differences in their anisotropies are obvious. While the anisotropy of the snow HDRF measured by Carlsen et al. (2020) is lower than that of the separated sea ice HDRF (the difference between both reduces to 0.12 at 59°), the anisotropy is significantly larger for the snow-covered sea ice HDRF (Goyens et al., 2018) with a maximum difference to the separated sea ice HDRF of 0.95 at 77°. In contrast to the other HDRF distributions with a minimum in nadir viewing direction, the minimum of the snow-covered sea ice HDRF by Goyens et al. (2018) is located in backward direction (at about −60°). The reasons for the anisotropy differences of both snow HDRFs (Carlsen et al., 2020, Goyens et al., 2018) remain unclear and might result from, e. g., snow grain size, impurity load or surface roughness. While the measurements from Carlsen et al. (2020) were at a wavelength 538 nm (green channel), the HDRF at 628 nm (red channel) by Goyens et al (2018) were used for comparison. However, the spectral dependence of the snow HDRF in the spectral range is small. The increased variability of the snow-covered sea ice HDRF observed by Goyens et al. (2018) might be due to the smaller footprint of the ground-based measurements compared to the airborne observations. In particular, small-scale surface roughness features can be resolved, which contribute to the variability of the ground-based measurements. In contrast to the snow-covered surfaces, the HDRF of bare ice (brown line) is significantly lower than the separated HDRF of the airborne observations and is characterized by an increased anisotropy. This is most prominent at reflection zenith angles of about 60°. However, the shape of the bare ice HDRF distribution is less smooth and shows a variability, that is even larger than that of the snow-covered sea ice HDRF. According to Goyens et al. (2018), this is due to the presence of thawed ice nearby highly reflective ice grains, which often occurs at the beginning of the melt season.**

**The magnitude of the separated sea ice HDRF analyzed in this study ranges between the literature values for snow-covered and bare ice HDRF. This is reasonable since the observed ice floes revealed a mixture of snow-covered and bare ice (e. g., Fig. 1c). The comparison of the different HDRFs illustrates the variability of the snow and sea ice HDRF in polar environments, which is affected by a variety of properties (e. g., snow cover, snow grain size or impurity concentration). Due to the significantly larger area fraction of sea ice compared to open ocean, the impact of the nearby darker open ocean surfaces in terms of horizontal photon transport should be much smaller (< 0.02) for sea ice and can, thus, not completely explain the differences between the analyzed HDRFs.**" (l. 302–328)

*327: open open*

The section where this occurred is revised anyway (see below).

*Figure 10: Why not include v_eff as legend in (d)? This would make the results clearer.*

Yes, good idea. We adjusted that. Note, that Fig. 10 is now Fig. 11, since we included a new histogram.

*355: I needed some double reading here until I understood, that 10d shows your main results for v_eff. Would be good to mention it already here.*

Also based on the comment on the same line from Referee #1, we tried to clarify that in the revised Section (see below).

*361: in in*

The section where this occurred is revised anyway (see below).

*361-365: It is difficult to follow you here, please consider rephrasing.*

With the comparison, we intended to assess the importance of the discrepancies of the open ocean HDRF (homogeneous vs. MIZ) on the constructed HDRF. The conclusion from this comparison is, that the impact of reduced wave intensity at the ocean surface in the MIZ is still obvious even for the constructed HDRF and needs to be considered when constructing a HDRF for the MIZ from individual single-surface HDRFs. We revised and rearranged Sect. 5 completely (see below). Hopefully, our intension is more clear from the revised Section (note that Equation 6 formerly was Eq. 5, and Fig. 11 was Fig. 10.):

[revised manuscript text omitted]

*398ff: This statement is difficult to prove, as the impact of the distribution was not evaluated in this study.*

That's true. We should formulate that a bit more carefully. Together with the previous sentence I would conclude that as follows: "**This approach could become relevant for randomly distributed sea ice and open ocean, where only the sea ice fraction is known.**" (l.417f). See also the revised conclusion section (below).

Revised conclusion:

"**Reflected radiance measurements were collected by an airborne 180° fish-eye camera in the MIZ north of Svalbard in June 2017. From these data, the HDRF was calculated during cloud-free conditions for a 20-minute sequence of 138 camera images covering different sea ice fractions. The HDRFs of sea ice and open ocean surfaces were separated by applying a sea ice mask with different reflectivity and color ratio thresholds.**

**From the separated images, the averaged HDRFs of open ocean and snow-covered sea ice surfaces in the MIZ were derived. They confirmed the general features of open ocean and sea ice reported in literature (e. g., Warren et al., 1998; Gatebe et al., 2003; Jackson and Alpers, 2010). However, a comparison with simulations indicated that the common BRDF parametrizations for homogeneous open ocean surfaces as function of the wind speed (Cox and Munk, 1954) partly differ in the MIZ. This is mainly due to wave attenuation between the ice floes in the MIZ (Kohout et al., 2011) leading to a reduced surface roughness compared to a homogeneous open ocean surface with the same surface wind speed. This effect narrows the sunglint and intensifies its magnitude. The irregular shape of the sunglint observed in the data set was a result of the limitation of the sunglint mask and the highly variable surface roughness associated with the irregular distribution of sea ice and open ocean in the MIZ.**

**The separated HDRF of partly snow-covered sea ice ranged between independent literature HDRFs of homogeneous snow and bare ice surfaces. However, the comparison also revealed the large diverstiy of snow/sea ice HDRF patterns associated with the variability of snow and ice properties. Minor differences between the HDRF in the MIZ and that of homogeneous surfaces could originate as a result of the radiative effects of the contrasting surface type nearby (e. g., Ricchiazzi et al. 1998, Schäfer et al., 2015).**

**The averaged HDRF of the MIZ showed features of both sea ice and open ocean surfaces. Even for rather high sea ice fractions, there is still a contribution from the sunglint in the MIZ, which might affect the analysis of satellite observations in these reflection angles. This especially holds regarding the permanent presence of leads in Arctic (e. g., Ivanova et al., 2016).**

**The mean MIZ HDRF of a subset of the analyzed data set was compared to the constructed one, calculated as a linear combination of the separated HDRFs of the remaining subset weighted by the sea ice fraction. The comparison showed good agreement for the measured sea ice fraction with a difference of less than 0.1 for 84 % of the pixels. Due to the assumption of a directionally constant sea ice fraction, the constructed HDRF of the MIZ was found to be smoother than the mean MIZ HDRF. Altogether, this analysis implies that the construction of the MIZ HDRF from individual sea ice and open ocean HDRFs provides meaningful results. This approach could become relevant for randomly distributed sea ice and open ocean, where only the sea ice fraction is known.**

**However, the impact of the wave attenuation on the open ocean HDRF in the MIZ has a significant impact also on the sunglint pattern of the MIZ HDRF. This effect needs to be considered in retrieval methods similar to the one used here. To improve the applicability of such methods, further**

research is needed, regarding the parametrization of the surface roughness of open ocean in the MIZ. Also the impact of the exact floe distribution on the surface reflectance properties needs to be investigated further. To extend the method to different environmental conditions (e. g., sea ice fraction, surface wind speed), further measurements are needed for a full parametrization of the HDRF in such complex scenarios as the MIZ, which may be the dominant surface type of the future Arctic."